# Quantifying the Inhibitory Efficacy of HIV-1 Therapeutic Interfering Particles at a Single CD4 T-Cell Resolution

**DOI:** 10.3390/v17101378

**Published:** 2025-10-15

**Authors:** Igor Sazonov, Dmitry Grebennikov, Rostislav Savinkov, Andreas Meyerhans, Gennady Bocharov

**Affiliations:** 1Faculty of Science and Engineering, Swansea University, Bay Campus, Swansea SA1 8EN, UK; 2Marchuk Institute of Numerical Mathematics of the RAS, 119333 Moscow, Russia; grebennikov_d_s@staff.sechenov.ru (D.G.); g.bocharov@inm.ras.ru (G.B.); 3Moscow Center of Fundamental and Applied Mathematics at INM RAS, 119333 Moscow, Russia; 4Institute for Computer Science and Mathematical Modelling, Sechenov First Moscow State Medical University, 119991 Moscow, Russia; 5Catalan Institution for Research and Advanced Studies, Pg. Lluis Companys 23, 08010 Barcelona, Spain; andreas.meyerhans@upf.edu; 6Infection Biology Laboratory, Universitat Pompeu Fabra, 08003 Barcelona, Spain

**Keywords:** HIV-1 life cycle, therapeutic interfering particles, mathematical models, CD4 T cell, inhibitory effect, single cell, reproduction efficiency

## Abstract

Efficient control of HIV-1 infection relies on highly active antiretroviral therapy (HAART). However, this therapy is not curative and requires continuous drug administration. Application of HIV-1 defective interfering particles (DIPs), engineered with ablations in key viral protein expressions (e.g., Tat, Rev, Vpu, and Env), suggests a therapeutic potential transforming them into Therapeutic Interfering Particles (TIPs). A recent animal HIV model study in non-human primates reports a substantial reduction in viral load after a single intravenous injection of TIPs. In contrast, human clinical trials demonstrate no beneficial effect of defective interfering particles (DIPs) in people living with HIV-1. This discrepancy highlights the importance of further investigation of HIV-TIP interactions. A quantitative view of intracellular replication for HIV-1 in the presence of TIPs is still missing. Here, we develop a high-resolution mathematical model to study various aspects of the interference of a specific engineered TIP-2 particle characterized by a 2.5-kb deletion in the HIV *pol-vpr* region with HIV-1 replication within infected CD4+ T cells. We define the conditions in terms of the number of homozygous HIV-1 virions and TIP-2 particles that enable the reduction of the wild-type virus replication number to the value of about one. The deterministic model predicts that at a ratio of 1 HIV-1 to 10 TIP-2 particles, the infected cell still produces some viruses, although in a minor quantity, i.e., about two virions per cycle. Pre-activation of the interferon type I (IFN-I) system results in a complete block of HIV-1 production by TIP-2 co-infected cells. Overall, the modelling results suggest that to improve the effectiveness of TIPs in reducing HIV infection, their combination with other types of antiviral protection should be considered. Our results can be used in the development of combination therapy aimed at treating HIV-1 infection.

## 1. Introduction

Human immunodeficiency virus type 1 (HIV-1) infection remains a major human health problem [1,2]. Efficient control of HIV-1 infection relies on lifelong use of highly active antiretroviral therapy (HAART) [3]. However, it is not curative and requires continuous drug administration. Other ways to control viral replication, which are under investigation, include the following:HIV vaccination [2];Preexposure prophylaxis [4];Combination therapies [5];Novel strategies such as CRISPR and CRISPR-based editing [6];Defective interfering particles (DIPs) which can be genetically engineered to become therapeutic interfering particles (TIPs) [7].

Modern genetic engineering makes it possible to create HIV-1 defective interfering particles (DIPs) by deleting key proteins: *tat*, *rev*, *vpu*, and *env* [8,9,10,11,12]. The successful use of the engineered DIPs in therapy allows them to be referred to as therapeutic interfering particles (TIPs). Animal HIV model studies in non-human primates show a substantial reduction of viral loads after a single intravenous injection of TIPs [8]. However, long-term investigation of two HIV-1-infected individuals revealed that (i) HIV-1 DIPs are present in vivo and (ii) superinfection with intact HIV-1 and DIPs takes place [9]. It is considered to be responsible for persistent viremia, systemic dissemination of defective proviruses, and accumulation of mutations in spite of HAART. Importantly, this interference does not affect wild-type viral replication and pathogenesis in vivo to a clinically significant extent [9]. The distinction between the structure of engineered TIPs and the naturally emerging DIPs in humans is an important factor contributing to the differing outcomes of the competition between HIV-1 and TIPs or DIPs in primate studies and in humans, respectively. The discrepancy between the two studies highlights the importance of further investigation of the HIV-TIP interaction.

HIV-1 is diploid, meaning that its viral genomic RNAs are packaged in pairs into the virion. Co-infection of a target cell with HIV-1 and DIPs potentially results in the generation of three types of virions: two homodimers, i.e., gRNA-HIV and gRNA-DIP, and the heterodimer gRNA-HIV/gRNA-DIP. An earlier in silico study [13] suggests that the competition between intact HIV-1 and DIPs would be evolutionarily unstable if the interference is associated with co-dimerization, i.e., generation of assorted or heterozygous genomes. In fact, it mirrors earlier empirical observations concerning the non-viability of heterozygous virions. Since heterodimerization enables recombination, there is a nontrivial safety risk, as HIV could recombine with TIP genomes, potentially restoring lost functions or generating new variants. Hence, the kinetics of heterozygous HIV-TIP generation requires further examination.

The aim of our study is to gain a quantitative understanding of the intracellular replication of HIV-1 in the presence of TIPs, in particular, the requirements for a substantial reduction of HIV progeny by TIPs. For this purpose, we have generated a mathematical model that describes the biochemical reactions during replication with HIV-1 and TIP-2 co-infected CD4+ T cells. The TIP-2 genomic structure is determined by a deletion of the trans-elements, i.e., a 2.5-kb deletion in the HIV *pol-vpr* region, and the reintroduction of the central polypurine tract (cPPT) as described in [8]. To study various aspects of the interference of TIPs with HIV-1 and the innate defense response (type I interferon, IFN-I) of the infected CD4+ T cell, we develop the mathematical model in deterministic and stochastic forms. We estimate the conditions, in terms of the multiplicity of infection (MOI) with HIV-1 and TIPs, that enable the reduction of the basic reproduction rate of the intact virus replication close to one, while keeping it slightly above one for TIPs.

In Section 2, we present the equations and parameter estimates of the deterministic model of cell superinfection with HIV-1 and TIPs. Section 3 deals with extensive numerical simulations with the model for various combinations of MOI and interferon abundance to quantify the inhibitory effect of TIPs. Implications for designing combination therapies aimed at curing HIV-1 infection are discussed in Section 4.

## 2. Materials and Methods

### 2.1. Biochemistry of HIV-1 and TIP Replication Cycle

The life cycle of HIV-1 and the induced antiviral type I interferon (IFN-I) response are described in Figure 1. We examine the replication of engineered TIP-2 particles that differ from HIV-1 by (i) a 2.5-kb deletion in the HIV *pol-vpr* region, (ii) ablation of the coding sequences of tat/rev/vpu/env, and (3) additional deletions in gag and env as detailed in [8]. Hence, the replication of TIP-2 follows the same stages except for those that are governed by the genome elements located within the 2.5-kb deleted region and the translation of proteins Tat, Rev, Vpu, Gag, and Env as presented in Figure 2. Based on this scheme, we develop a parallel set of equations for the TIP-2 replication in a single CD4+ T cell. Notice that both HIV-1 and TIP-2 induce the antiviral response elements, resulting in IFN-I synthesis and IFN-receptor-transduced responses of interferon-stimulated genes (ISGs) in the infected cell. Co-infection of target cells with HIV-1 and TIP-2 results in production of (i) infectious virions with gRNA-HIV, (ii) defective particles containing gRNA-TIP-2, and (iii) the heterozygous particles with gRNA-HIV/gRNA-TIP. The biochemical scheme illustrating the formation of homozygous HIV-1 virions, homozygous TIP-2 particles, and the heterozygous HIV-TIP-2 particles is shown in Figure 3. As follows from the replication cycle of HIV-1 and TIP-2 and their genomic structures (see Figure 1 and Figure 2), the model assumes that the viruses and TIPs compete for the viral proteins Gag-Pol, Gag, and gp-160. The set of equations below for the individual reactions is specified using basic principles of biochemical kinetics. We expand on our previous work [14]. Parameters of the biochemical reactions underlying the HIV-1 and TIP-2 replication stages are assumed to be the same, except for the rates of RNA transcription and splicing, which are corrected for the shorter length of TIP-2. To describe the rate-limiting reactions representing the Tat- and Rev-dependent regulation of mRNA abundance, a Michaelis–Menten-type parameterization is used. To describe the assembly of pre-virions, pre-TIP-2, and pre-HIV-TIP-2 particles, we assume that the dependence of the assembly rate on Gag-Pol, Gag, and gp-160 saturates at high levels of these proteins. The dependence of the assembly rate on the number of RNA molecules follows second-order kinetics due to the diploid nature of the viral genome but saturates at high densities of the genomic RNAs.

### 2.2. Deterministic Equations for HIV-1 and TIP-2

**Cell entry:** Free viral particles $\left[V_{\mathrm{free}}\right]$ and TIP-2 particles $\left[{\mathrm{TIP}}_{\mathrm{free}}\right]$ are bound to the cell membrane according to the following equations: (1)$$\begin{array}{cc} {\dot{x}}_{1} & =-k_{\mathrm{bound}}x_{1}-d_{\mathrm{free}}x_{1} \end{array}$$(2)$$\begin{array}{cc} \;{\dot{x}}_{1T} & =-k_{\mathrm{bound}}x_{1T}-d_{\mathrm{free}}x_{1T} \end{array}$$(3)$$\begin{array}{cc} {\dot{x}}_{2} & =-k_{\mathrm{bound}}x_{1}-k_{\mathrm{fuse}}x_{2}-d_{\mathrm{bound}}x_{2} \end{array}$$(4)$$\begin{array}{cc} {\dot{x}}_{2T} & =-k_{\mathrm{bound}}x_{1T}-k_{\mathrm{fuse}}x_{2T}-d_{\mathrm{bound}}x_{2T} \end{array}$$
Here, the following notation is used:$x_{1}=\left[V_{\mathrm{free}}\right]$ is the number of free HIV-1 virions in the vicinity of the cell;$x_{1T}=\left[{\mathrm{TIP}}_{\mathrm{free}}\right]$ is the number of free TIPs in the vicinity of the cell;$x_{2}=\left[V_{\mathrm{bound}}\right]$ is the number of virions;$x_{2T}=\left[{\mathrm{TIP}}_{\mathrm{bound}}\right]$ is the number of TIPs bound to CD4 and the co-receptor.

The parameter values are set according to [14], $k_{\mathrm{bound}}=3.1$ $h^{-1}$, $k_{\mathrm{fuse}}=0.7$ $h^{-1}$, $d_{\mathrm{free}}=0.38$ $h^{-1}$, $d_{\mathrm{bound}}=0.0008$ $h^{-1}$.

**RNA release, reverse transcription, and integration:** Following the entry of HIV-1 and TIP-2 into the cell, the release of RNA takes place, and reverse transcription is initiated, resulting in the production of proviral DNA. The antiviral response of the cell results in the production of proteins SAMHD1 and APOBEC3, which exert an inhibitory effect on this initial step in viral/TIP-2 replication. The genomic DNA penetrates the nucleus and is integrated into the chromosomal DNA of the host cell. Accordingly, the abundance of the following components is considered:$x_{3}=\left[{\mathrm{RNA}}_{\mathrm{cor}}\right]$ is the number of genomic RNA molecules in the cytoplasm;$x_{3T}=\left[{\mathrm{RNA}}_{\mathrm{cor}}^{\mathrm{TIP}}\right]$ is the number of TIP RNA molecules in the cytoplasm;$x_{4}=\left[{\mathrm{DNA}}_{\mathrm{cor}}\right]$ is the number of proviral DNA molecules synthesised by reverse transcription;$x_{4T}=\left[{\mathrm{DNA}}_{\mathrm{cor}}^{\mathrm{TIP}}\right]$ is the number of proviral TIP DNA molecules synthesised by reverse transcription;$x_{5}=\left[{\mathrm{DNA}}_{\mathrm{nuc}}\right]$ is the number of DNA molecules in the nucleus;$x_{5T}=\left[{\mathrm{DNA}}_{\mathrm{nuc}}^{\mathrm{TIP}}\right]$ is the number of TIP DNA molecules in the nucleus;$x_{6}=\left[{\mathrm{DNA}}_{\mathrm{int}}\right]$ is the number of integrated DNA;$x_{6T}=\left[{\mathrm{DNA}}_{\mathrm{int}}^{\mathrm{TIP}}\right]$ is the number of integrated  TIP DNA;$x_{34}=\left[\mathrm{APOBEC}3\right]$ is the number of molecules of the apolipoprotein B editing complex (APOBEC3) induced due to HIV-1 and TIPs;$x_{35}=\left[\mathrm{SAMHD}1\right]$ is the number of molecules of the cellular enzyme (SAMHD1), responsible for blocking the replication of HIV-1 in resting CD4+ T lymphocytes, induced by HIV-1 and TIP-2.

The corresponding equations are(5)$$\begin{array}{cc} {\dot{x}}_{3} & =k_{\mathrm{fuse}}x_{2}-k_{\mathrm{RT}}x_{3}-d_{{\mathrm{RNA}}_{\mathrm{cor}}}x_{3}-f_{\mathrm{APO}}x_{34}x_{3}-f_{\mathrm{SAM}}x_{35}x_{3} \end{array}$$(6)$$\begin{array}{cc} {\dot{x}}_{3T} & =k_{\mathrm{fuse}}x_{2T}-k_{\mathrm{RT}}^{\mathrm{TIP}}x_{3T}-d_{{\mathrm{RNA}}_{\mathrm{cor}}}x_{3T}-f_{\mathrm{APO}}x_{34}x_{3T}-f_{\mathrm{SAM}}x_{35}x_{3T} \end{array}$$(7)$$\begin{array}{cc} {\dot{x}}_{4} & =k_{\mathrm{RT}}x_{3}-k_{{\mathrm{DNA}}_{t}}x_{4}-d_{{\mathrm{DNA}}_{\mathrm{cor}}}x_{4} \end{array}$$(8)$$\begin{array}{cc} {\dot{x}}_{4T} & =k_{\mathrm{RT}}^{\mathrm{TIP}}x_{3T}-k_{{\mathrm{DNA}}_{t}}x_{4T}-d_{{\mathrm{DNA}}_{\mathrm{cor}}}x_{4T} \end{array}$$(9)$$\begin{array}{cc} {\dot{x}}_{5} & =k_{{\mathrm{DNA}}_{t}}x_{4}-k_{\mathrm{int}}x_{5}-d_{{\mathrm{DNA}}_{\mathrm{nuc}}}x_{5} \end{array}$$(10)$$\begin{array}{cc} {\dot{x}}_{5T} & =k_{{\mathrm{DNA}}_{t}}x_{4T}-k_{\mathrm{int}}x_{5T}-d_{{\mathrm{DNA}}_{\mathrm{nuc}}}x_{5T} \end{array}$$(11)$$\begin{array}{cc} {\dot{x}}_{6} & =k_{\mathrm{int}}x_{5}-d_{{\mathrm{DNA}}_{\mathrm{int}}}x_{6} \end{array}$$(12)$$\begin{array}{cc} {\dot{x}}_{6T} & =k_{\mathrm{int}}x_{5T}-d_{{\mathrm{DNA}}_{\mathrm{int}}}x_{6T} \end{array}$$
with the following estimates of the parameters: $k_{\mathrm{RT}}=0.43$ $h^{-1}$, $k_{\mathrm{RT}}^{\mathrm{TIP}}=0.57$ $h^{-1}$ (taking into account 25% shorter genome), $k_{{\mathrm{DNA}}_{t}}=0.12$ $h^{-1}$, $k_{\mathrm{int}}=0.14$ $h^{-1}$, $d_{{\mathrm{RNA}}_{\mathrm{cor}}}=0.21$ $h^{-1}$, $d_{{\mathrm{DNA}}_{\mathrm{cor}}}=0.03$ $h^{-1}$, $d_{{\mathrm{DNA}}_{\mathrm{nuc}}}=0.001$ $h^{-1}$; $d_{{\mathrm{DNA}}_{\mathrm{int}}}=0.00002$ $h^{-1}$, $f_{\mathrm{APO}}=0.35$ $h^{-1}$, $f_{\mathrm{SAM}}=1.6$ $h^{-1}$.

Note that the last two terms in Equations (Equation 5) and (Equation 6) describe the inhibitory effect of the SAMHD1 and APOBEC3 proteins produced in response to HIV-1 and TIP-2 genomes.

**Transcription and splicing:** RNA transcription and splicing take place in the nucleus. For HIV-1, the generation of full-length (mRNAg), single-spliced (mRNAss), and double-spliced (mRNAds) messenger RNA molecules takes place. The RNAs are then transported to the cytoplasm (mRNACg, mRNACss, mRNACds). In the case of TIP-2, only the transcription of full-length (TIP mRNAg) takes place. The difference in genomic organization of HIV and TIP-2 is revealed at the RNA transcription stage. The Tat-dependent regulation of the transcription rate takes into account that TIP-2 has a shorter nucleotide length. The abundance of the above components is characterized by the following state variables:$x_{7}=\left[{\mathrm{mRNA}}_{g}\right]$ is the number of HIV mRNA molecules in the nucleus: g for genomic or full-length;$x_{7T}=\left[{\mathrm{mRNA}}_{g}^{\mathrm{TIP}}\right]$ is the number of TIP-2 mRNA molecules in the nucleus: g for genomic or full-length;$x_{8}=\left[{\mathrm{mRNA}}_{\mathrm{ss}}\right]$ is the number of HIV singly spliced (ss) mRNA molecules in the nucleus;$x_{9}=\left[{\mathrm{mRNA}}_{\mathrm{ds}}\right]$ is the number of HIV doubly spliced (ds) mRNA molecules in the nucleus;$x_{10}=\left[{\mathrm{mRNAc}}_{g}\right]$ is the number of HIV mRNA molecules in the cytoplasm: g for genomic or full-length;$x_{10T}=\left[{\mathrm{mRNAc}}_{g}^{\mathrm{TIP}}\right]$ is the number of TIP-2 mRNA molecules in the cytoplasm: g for genomic;$x_{11}=\left[{\mathrm{mRNAc}}_{\mathrm{ss}}\right]$ is the number of HIV singly spliced (ss) mRNA molecules in the cytoplasm;$x_{12}=\left[{\mathrm{mRNAc}}_{\mathrm{ds}}\right]$ is the number of HIV doubly spliced (ds) mRNA molecules in the cytoplasm.

The model equations for the respective biochemical reactions are specified as listed below: (13)$$\begin{array}{cc} {\dot{x}}_{7} & =f_{\mathrm{TR}}\;x_{6}-k_{{\mathrm{ssRNA}}_{g}}g_{\mathrm{Rev}}x_{7}-k_{{\mathrm{eRNA}}_{g}}f_{\mathrm{Rev}}x_{7}-d_{{\mathrm{RNA}}_{g}}x_{7} \end{array}$$(14)$$\begin{array}{cc} {\dot{x}}_{7T} & =f_{\mathrm{TR}}^{\mathrm{TIP}}\;x_{6T}-k_{{\mathrm{eRNA}}_{g}}f_{\mathrm{Rev}}x_{7T}-d_{{\mathrm{RNA}}_{g}}x_{7T} \end{array}$$(15)$$\begin{array}{cc} {\dot{x}}_{8} & =k_{{\mathrm{ssRNA}}_{g}}g_{\mathrm{Rev}}\;x_{7}-k_{{\mathrm{dsRNA}}_{\mathrm{ss}}}g_{\mathrm{Rev}}x_{8}-k_{{\mathrm{eRNA}}_{\mathrm{ss}}}f_{\mathrm{Rev}}x_{8}-d_{{\mathrm{RNA}}_{\mathrm{ss}}}x_{8} \end{array}$$(16)$$\begin{array}{cc} {\dot{x}}_{9} & =k_{{\mathrm{dsRNA}}_{\mathrm{ss}}}g_{\mathrm{Rev}}\;x_{8}-k_{{\mathrm{eRNA}}_{\mathrm{ds}}}x_{9}-d_{{\mathrm{RNA}}_{\mathrm{ds}}}x_{9} \end{array}$$(17)$$\begin{array}{cc} {\dot{x}}_{10} & =k_{{\mathrm{eRNA}}_{g}}f_{\mathrm{Rev}}\;x_{7}-k_{\mathrm{tp},\mathrm{RNA}}x_{10}-d_{{\mathrm{RNA}}_{g}}x_{10} \end{array}$$(18)$$\begin{array}{cc} {\dot{x}}_{10T} & =k_{{\mathrm{eRNA}}_{g}}f_{\mathrm{Rev}}\;x_{7T}-k_{\mathrm{tp},\mathrm{RNA}}x_{10T}-d_{{\mathrm{RNA}}_{g}}x_{10T} \end{array}$$(19)$$\begin{array}{cc} {\dot{x}}_{11} & =k_{{\mathrm{eRNA}}_{\mathrm{ss}}}f_{\mathrm{Rev}}\;x_{8}-d_{{\mathrm{RNA}}_{\mathrm{ss}}}x_{11} \end{array}$$(20)$$\begin{array}{cc} {\dot{x}}_{12} & =k_{{\mathrm{eRNA}}_{\mathrm{ds}}}x_{9}-d_{{\mathrm{RNA}}_{\mathrm{ds}}}x_{12}. \end{array}$$

The parameterized functions $f_{\mathrm{TR}}\left(\cdot \right)$, $f_{\mathrm{Rev}}\left(\cdot \right)$, and $g_{\mathrm{Rev}}\left(\cdot \right)$ describe the Tat-dependent regulation of transcription and Rev-dependent regulation of export from nucleus to cytoplasm:(21)$$f_{\mathrm{TR}}={\mathrm{TR}}_{\mathrm{cell}}+\frac{x_{13}}{\theta_{\mathrm{Tat}}+x_{13}}{\mathrm{TR}}_{\mathrm{Tat}},\;f_{\mathrm{TR}}^{\mathrm{TIP}}=1.33f_{\mathrm{TR}}.$$(22)$$f_{\mathrm{Rev}}=\frac{x_{14}}{\theta_{\mathrm{Rev}}+x_{14}},\;g_{\mathrm{Rev}}=1-\beta f_{\mathrm{Rev}},$$
where $x_{13}=\left[\mathrm{Tat}\right]$, $x_{14}=\left[\mathrm{Rev}\right]$ are abundances of Tat and Rev protein molecules. In the estimation for $f_{\mathrm{TR}}^{\mathrm{TIP}}$, it is taken into account that TIP-2 has a shorter nucleotide length by 2.5 kb because of eliminating *tat*/*rev*/*vpu*/*env*.

The following estimates for the reaction rate constants are used $k_{{\mathrm{ssRNA}}_{g}}=k_{{\mathrm{dsRNA}}_{\mathrm{ss}}}=2.4$ $h^{-1}$, $k_{{\mathrm{eRNA}}_{g}}=k_{{\mathrm{eRNA}}_{\mathrm{ss}}}=2.3$ $h^{-1}$, $k_{{\mathrm{eRNA}}_{\mathrm{ds}}}=4.6$ $h^{-1}$, $k_{\mathrm{tp},\mathrm{RNA}}=2.8$ $h^{-1}$, $d_{{\mathrm{RNA}}_{g}}=d_{{\mathrm{RNA}}_{\mathrm{ss}}}=d_{{\mathrm{RNAc}}_{\mathrm{ds}}}=0.12$ $h^{-1}$, ${\mathrm{TR}}_{\mathrm{cell}}=15$ $h^{-1}$, ${\mathrm{TR}}_{\mathrm{Tat}}=1500$ $h^{-1}$, $\theta_{\mathrm{Tat}}=10^{3}$ molecules, $\theta_{\mathrm{Rev}}=7.7\times 10^{4}$ molecules, β=0.9.

**Protein translation:** In the HIV-1 life cycle, the encoded messenger RNAs are translated by the ribosomes to produce the proteins Tat, Rev, Gag-Pol, Gag, and gp160 required for the regulation and assembly of new virions, and the regulatory proteins Vpu and Vif affecting the interferon system response. In the case of the TIP-2 life cycle, the encoded messenger RNAs result in the production of the Gag proteins required for the assembly of new TIP-2 particles. The abundance of Gag-Pol, Gag, gp160, Vpu, and Vif protein molecules is characterized by the following model variables: $x_{15}=\left[\mathrm{Gag}-\mathrm{Pol}\right]$; $x_{16}=\left[\mathrm{Gag}\right]$, $x_{16T}=\left[{\mathrm{Gag}}^{\mathrm{TIP}}\right]$, $x_{17}=\left[\mathrm{gp}160\right]$; $x_{18}=\left[\mathrm{Vpu}\right]$; $x_{19}=\left[\mathrm{Vif}\right]$. The respective model equations are as follows: (23)$$\begin{array}{cc} {\dot{x}}_{13} & =k_{\mathrm{trans}}f_{\mathrm{ds},\mathrm{Tat}}x_{12}-d_{\mathrm{TaT}}x_{13} \end{array}$$(24)$$\begin{array}{cc} {\dot{x}}_{14} & =k_{\mathrm{trans}}f_{\mathrm{ds},\mathrm{Rev}}x_{12}-d_{\mathrm{Rev}}x_{14} \end{array}$$(25)$$\begin{array}{cc} {\dot{x}}_{15} & =k_{\mathrm{trans}}f_{g,\mathrm{Gag}-\mathrm{Pol}}x_{10}-k_{\mathrm{tp},\mathrm{Gag}-\mathrm{Pol}}x_{15}-d_{\mathrm{Gag}-\mathrm{Pol}}x_{15} \end{array}$$(26)$$\begin{array}{cc} {\dot{x}}_{16} & =k_{\mathrm{trans}}f_{g,\mathrm{Gag}}x_{10}-k_{\mathrm{tp},\mathrm{Gag}}x_{16}-d_{\mathrm{Gag}}x_{16} \end{array}$$(27)$$\begin{array}{cc} {\dot{x}}_{16T} & =k_{\mathrm{trans}}f_{g,\mathrm{Gag}}x_{10T}-k_{\mathrm{tp},\mathrm{Gag}}x_{16T}-d_{\mathrm{Gag}}x_{16T} \end{array}$$(28)$$\begin{array}{cc} {\dot{x}}_{17} & =k_{\mathrm{trans}}f_{\mathrm{gp}160}x_{11}-k_{\mathrm{tp},\mathrm{gp}160}x_{17}-d_{\mathrm{gp}160}x_{17} \end{array}$$(29)$$\begin{array}{cc} {\dot{x}}_{18} & =k_{\mathrm{trans}}f_{\mathrm{ss},\mathrm{Vpu}}x_{11}-k_{\mathrm{tp},\mathrm{Vpu}}x_{18}-d_{\mathrm{Vpu}}x_{18} \end{array}$$(30)$$\begin{array}{cc} {\dot{x}}_{19} & =k_{\mathrm{trans}}f_{\mathrm{ss},\mathrm{Vif}}x_{11}-d_{\mathrm{Vif}}x_{19} \end{array}$$
Here, the following parameter estimates are used: $f_{g,\mathrm{Gag}-\mathrm{Pol}}=0.05$, $f_{g,\mathrm{Gag}}=0.95$, $f_{\mathrm{ss},\mathrm{gp}160}=0.64$, $f_{\mathrm{ds},\mathrm{Tat}}=0.025$, $f_{\mathrm{ds},\mathrm{Rev}}=0.2$, $f_{\mathrm{ss},\mathrm{Vpu}}=0.062$, $f_{\mathrm{ss},\mathrm{Vif}}=0.145$, $k_{\mathrm{trans}}=524$ $h^{-1}$, $k_{\mathrm{tp},\mathrm{Gag}-\mathrm{Pol}}=k_{\mathrm{tp},\mathrm{Gag}}=k_{\mathrm{tp},\mathrm{gp}160}=k_{\mathrm{tp},\mathrm{Vpu}}=2.8$ $h^{-1}$, $d_{\mathrm{Gag}-\mathrm{Pol}}=d_{\mathrm{Gag}}=0.09$ $h^{-1}$, $d_{\mathrm{gp}160}=0.02$ $h^{-1}$, $d_{\mathrm{Tat}}=0.04$ $h^{-1}$, $d_{\mathrm{Rev}}=0.07$ $h^{-1}$, $d_{\mathrm{Vpu}}=0.39$ $h^{-1}$, $d_{\mathrm{Vif}}=1.38$ $h^{-1}$.

**Assembly of pre-virions, pre-TIP-2, and pre-HIV-TIP-2 particles:** Viral proteins Gag-Pol, Gag, and gp160 and viral/TIP-2 RNA molecules are transported to the cell membrane where they form homozygous pre-HIV-1 virions, homozygous pre-TIP-2 particles, and heterozygous pre-HIV-TIP-2 particles. The Vpu protein is also moved to the membrane. The model variables characterizing the abundance of the above components are as follows: $x_{20}=\left[{\mathrm{Gag}-\mathrm{Pol}}_{\mathrm{mem}}\right]$, $x_{21}=\left[{\mathrm{Gag}}_{\mathrm{mem}}\right]$, $x_{22}=\left[{\mathrm{gp}160}_{\mathrm{mem}}\right]$, are, respectively, the number of the Gag-Pol, Gag, and gp160 viral/TIP-2 protein molecules at the membrane;
$x_{23}=\left[{\mathrm{RNA}}_{\mathrm{mem}}\right]$ is the number of viral RNA molecules at the membrane;$x_{23T}=\left[{\mathrm{RNA}}_{\mathrm{mem}}^{\mathrm{TIP}}\right]$ is the number of TIP-2 RNA molecules at the membrane;$x_{24}=\left[{\mathrm{Vpu}}_{\mathrm{mem}}\right]$ is the number of Vpu protein molecules at the membrane;$x_{25}=\left[\mathrm{pre}-\mathrm{Virion}\right]$ is the number of pre-virion complexes at the membrane (containing two HIV RNA molecules);$x_{25T}=\left[\mathrm{pre}-\mathrm{TIP}\right]$ is the number of pre-TIP-2 complexes at the membrane (containing two TIP RNA molecules);$x_{25\mathrm{HT}}=\left[\mathrm{pre}-\mathrm{HIV}-\mathrm{TIP}\right]$ is the number of heterozygous pre-HIV-TIP-2 complexes at the membrane (containing one HIV RNA and one TIP-2 RNA).

We describe this stage using the following equations: (31)$$\begin{array}{cc} {\dot{x}}_{20} & =k_{\mathrm{tp},\mathrm{Gag}-\mathrm{Pol}}x_{15}-k_{\mathrm{comb}}N_{\mathrm{Gag}-\mathrm{Pol}}\left(F_{c}\;\theta +\;F_{c}^{TIP}\;\theta_{T}+\;F_{c}^{HT}\;\theta_{\mathrm{HT}}\right)-d_{\mathrm{mem},\mathrm{Gag}-\mathrm{Pol}}x_{20} \end{array}$$(32)$$\begin{array}{cc} {\dot{x}}_{21} & =k_{\mathrm{tp},\mathrm{Gag}}\left(x_{16}+x_{16T}\right)-k_{\mathrm{comb}}N_{\mathrm{Gag}}\left(F_{c}\;\theta +\;F_{c}^{TIP}\;\theta_{T}+\;F_{c}^{HT}\;\theta_{\mathrm{HT}}\right)-d_{\mathrm{mem},\mathrm{Gag}}x_{21} \end{array}$$(33)$$\begin{array}{cc} {\dot{x}}_{22} & =k_{\mathrm{tp},\mathrm{gp}160}x_{17}-k_{\mathrm{comb}}N_{\mathrm{gp}160}\left(F_{c}\;\theta +\;F_{c}^{TIP}\;\theta_{T}+\;F_{c}^{HT}\;\theta_{\mathrm{HT}}\right)-d_{\mathrm{mem},\mathrm{gp}160}x_{22} \end{array}$$(34)$$\begin{array}{cc} {\dot{x}}_{23} & =k_{\mathrm{tp},\mathrm{RNA}}x_{10}-k_{\mathrm{comb}}N_{\mathrm{RNA}}\;F_{c}\;\theta -k_{\mathrm{comb}}\;F_{c}^{HT}\;\theta_{\mathrm{HT}}-d_{{\mathrm{RNA}}_{g}}x_{23} \end{array}$$(35)$$\begin{array}{cc} {\dot{x}}_{23T} & =k_{\mathrm{tp},\mathrm{RNA}}x_{10T}-k_{\mathrm{comb}}N_{\mathrm{RNA}}\;F_{c}^{TIP}\;\theta_{T}-k_{\mathrm{comb}}\;F_{c}^{HT}\;\theta_{\mathrm{HT}}-d_{{\mathrm{RNA}}_{g}}x_{23T} \end{array}$$(36)$$\begin{array}{cc} {\dot{x}}_{24} & =k_{\mathrm{tp},\mathrm{Vpu}}x_{18}-d_{\mathrm{Vpu}}x_{24} \end{array}$$(37)$$\begin{array}{cc} {\dot{x}}_{25} & =k_{\mathrm{comb}}\;F_{c}\;\theta -k_{\mathrm{bud}}x_{25}-d_{\mathrm{comb}}x_{25} \end{array}$$(38)$$\begin{array}{cc} {\dot{x}}_{25T} & =k_{\mathrm{comb}}\;F_{c}^{TIP}\;\theta_{T}-k_{\mathrm{bud}}x_{25T}-d_{\mathrm{comb}}x_{25T} \end{array}$$(39)$$\begin{array}{cc} {\dot{x}}_{25\mathrm{HT}} & =k_{\mathrm{comb}}\;F_{c}^{HT}\;\theta_{\mathrm{HT}}-k_{\mathrm{bud}}x_{25\mathrm{HT}}-d_{\mathrm{comb}}x_{25\mathrm{HT}} \end{array}$$
with the following parameterizations of the assembly process:(40)$$F_{c}=\frac{x_{20}}{x_{20}+K_{V_{\mathrm{rel}}}N_{\mathrm{Gag}-\mathrm{Pol}}}\cdot \frac{x_{21}}{x_{21}+K_{V_{\mathrm{rel}}}N_{\mathrm{Gag}}}\cdot \frac{x_{22}}{x_{22}+K_{V_{\mathrm{rel}}}N_{\mathrm{gp}160}},\;K_{V_{\mathrm{rel}}}=10^{3},$$(41)$$F_{c}^{\mathrm{TIP}}=F_{c}^{HT}=F_{c},$$(42)$$\theta =\frac{x_{23}^{2}}{1+x_{23}},\;\theta_{T}=\frac{x_{23T}^{2}}{1+x_{23T}},\;\theta_{HT}=\frac{x_{23}x_{23T}}{1+x_{23}+x_{23T}}.$$
Here, we take $k_{\mathrm{comb}}=8.0$ $h^{-1}$, $k_{\mathrm{bud}}=2.0$ $h^{-1}$, $k_{\mathrm{tp},\mathrm{RNA}}=k_{\mathrm{tp},\mathrm{acc}}=k_{\mathrm{tp}}=2.8$ $h^{-1}$, $N_{\mathrm{RNA}}=2$, $N_{\mathrm{Gag}-\mathrm{Pol}}=250$, $N_{\mathrm{Gag}}=5000$, $N_{\mathrm{gp}160}=24$, $d_{{\mathrm{RNA}}_{g}}=0.12$ $h^{-1}$, $d_{\mathrm{mem},\mathrm{Gag}-\mathrm{Pol}}=d_{\mathrm{mem},\mathrm{Gag}}=0.004$ $h^{-1}$, $d_{\mathrm{mem},\mathrm{gp}160}=0.014$ $h^{-1}$, $d_{\mathrm{comb}}=0.52$ $h^{-1}$.

**Budding and release of HIV-1 virions and TIP-2 particles:** The assembled HIV-1 virions, TIP-2, and HIV-1-TIP-2 particles are secreted, leaving the cell membrane. The release is suppressed by the interferon-stimulated Tetherin protein. However, the viral Vpu protein located at the membrane blocks the Tetherin molecules. The abundance of the respective components is described by the following set of state variables:$x_{26}=\left[V_{\mathrm{bud}}\right]$ is the number of immature HIV-1 virions after budding from the cell;$x_{26T}=\left[{\mathrm{TIP}}_{\mathrm{bud}}\right]$ is the number of immature TIP-2 particles after budding from the cell;$x_{26\mathrm{HT}}=\left[{\mathrm{HIV}-\mathrm{TIP}}_{\mathrm{bud}}\right]$ is the number of immature HIV-TIP particles after budding from the cell;$x_{27}=\left[V_{\mathrm{mat}}\right]$ is the number of mature HIV-1 virions outside the cell;$x_{27T}=\left[{\mathrm{TIP}}_{\mathrm{mat}}\right]$ is the number of mature TIP-2 particles outside the cell;$x_{27\mathrm{HT}}=\left[{\mathrm{HIV}-\mathrm{TIP}}_{\mathrm{mat}}\right]$ is the number of mature HIV-TIP particles outside the cell;$x_{36}=\left[\mathrm{Tetherin}\right]$ is the number of Tetherin molecules;

Their dynamics are described by equations: (43)$$\begin{array}{cc} {\dot{x}}_{26} & =k_{\mathrm{bud}}x_{25}-k_{\mathrm{mat}}x_{26}-f_{\mathrm{Teth}}x_{36}x_{26}-d_{\mathrm{bud}}x_{26} \end{array}$$(44)$$\begin{array}{cc} {\dot{x}}_{26T} & =k_{\mathrm{bud}}x_{25T}-k_{\mathrm{mat}}x_{26T}-f_{\mathrm{Teth}}x_{36}x_{26T}-d_{\mathrm{bud}}x_{26T} \end{array}$$(45)$$\begin{array}{cc} {\dot{x}}_{26\mathrm{HT}} & =k_{\mathrm{bud}}x_{25\mathrm{HT}}-k_{\mathrm{mat}}x_{26\mathrm{HT}}-f_{\mathrm{Teth}}x_{36}x_{26\mathrm{HT}}-d_{\mathrm{bud}}x_{26\mathrm{HT}} \end{array}$$(46)$$\begin{array}{cc} {\dot{x}}_{27} & =k_{\mathrm{mat}}x_{26}-d_{\mathrm{free}}x_{27}; \end{array}$$(47)$$\begin{array}{cc} {\dot{x}}_{27T} & =k_{\mathrm{mat}}x_{26T}-d_{\mathrm{free}}x_{27T}; \end{array}$$(48)$$\begin{array}{cc} {\dot{x}}_{27\mathrm{HT}} & =k_{\mathrm{mat}}x_{26\mathrm{HT}}-d_{\mathrm{free}}x_{27\mathrm{HT}}. \end{array}$$
Here, $k_{\mathrm{mat}}=2.4$ $h^{-1}$, $d_{\mathrm{bud}}=0.38$ $h^{-1}$, $f_{\mathrm{Teth}}=0.008$ $h^{-1}$.

**Activation of the IFN-I system in infected cells:** The appearance of RNAs following the entry of HIV-1 and TIP-2 induces the antiviral response elements, resulting in IFN-I synthesis and IFN-receptor-transduced responses of ISGs in the infected cell. In particular, the RIG-1 protein is activated, initiating the production of NF-kB and IRF3 proteins, which finally induce the synthesis of intracellular IFN-I. The respective state variables and equations of the model read as follows:$x_{28}=\left[\mathrm{RIG}1\right]$ is the number of the RIG-1 protein molecules activated in response to HIV-1 and TIP-2 RNAs;$x_{29}=\left[\mathrm{IRF}3\right]$ s the numbers of the IRF3 proteins$x_{30}=\left[\mathrm{NF}-\mathrm{kb}\right]$ is the numbers of the NF-kb proteins;$x_{31}=\left[\mathrm{IFNi}\right]$ is the number of intracellular IFN-I molecules.(49)$$\begin{array}{cc} {\dot{x}}_{28} & =k_{\mathrm{RIG}1}\left(x_{3}+x_{3T}\right)-d_{\mathrm{RIG}}x_{28} \end{array}$$(50)$$\begin{array}{cc} {\dot{x}}_{29} & =k_{\mathrm{IRF}3}x_{28}-d_{\mathrm{IRF}3}x_{29} \end{array}$$(51)$$\begin{array}{cc} {\dot{x}}_{30} & =k_{\mathrm{NF}-\mathrm{kB}}x_{28}-d_{\mathrm{NF}-\mathrm{kB}}x_{30} \end{array}$$(52)$$\begin{array}{cc} {\dot{x}}_{31} & =k_{\mathrm{IFNi}}x_{29}+k_{\mathrm{IFNi}}x_{30}-k_{e}x_{31}-d_{\mathrm{IFNi}}x_{31}. \end{array}$$
Here, we use the following parameter estimates from [14]: $k_{\mathrm{RIG}1}=0.48$ $h^{-1}$, $k_{\mathrm{IRF}3}=1.02$ $h^{-1}$, $k_{\mathrm{IFNi}}=1$ $h^{-1}$, $k_{e}=0.13$ $h^{-1}$, $k_{\mathrm{NF}-\mathrm{kB}}=0.91$ $h^{-1}$, $d_{\mathrm{RIG}}=0.4$ $h^{-1}$, $d_{\mathrm{IFNi}}=0.08$ $h^{-1}$, $d_{\mathrm{IRF}3}=0.0015$ $h^{-1}$, $d_{\mathrm{NF}-\mathrm{kB}}=0.00026$ $h^{-1}$.

**IFN-I stimulated protein production:** Synthesized IFN-I molecules exit the cell ($\left[{IFN}_{e}\right]$). The extracellular interferon can bind to the IFN receptors on the membrane of the CD4+ T cell, activating the proteins STAT1 and 2, which start the production of the virus replication inhibitor proteins (APOBEC3, SAMHD1, and Tetherin). The respective model variables are as follows:$x_{32}=\left[\mathrm{IFNe}\right]$ is the number of extracellular IFN-I molecules;$x_{33}=\left[\mathrm{STAT}1,2\right]$ is the number of STAT-1–STAT-2 heterodimers;$x_{34}=\left[\mathrm{APOBEC}3\right]$ is the number of molecules of the apolipoprotein B editing complex;$x_{35}=\left[\mathrm{SAMHD}1\right]$ is the number of molecules of a cellular enzyme responsible for blocking the replication of HIV in resting CD4+ T lymphocytes;$x_{36}=\left[\mathrm{Tetherin}\right]$ is the number of Tetherin molecules.

The equations for the dynamics of extracellular IFN-I, APOBEC3, SAMHD1, and Tetherin have the form: (53)$$\begin{array}{cc} {\dot{x}}_{32} & =k_{e}x_{31}-d_{\mathrm{IFNe}}x_{32} \end{array}$$(54)$$\begin{array}{cc} {\dot{x}}_{33} & =k_{\mathrm{STAT}}x_{32}-d_{\mathrm{STAT}}x_{33} \end{array}$$(55)$$\begin{array}{cc} {\dot{x}}_{34} & =k_{\mathrm{ISG}}x_{33}-f_{\mathrm{Vif}}x_{19}x_{34}-d_{\mathrm{APO}}x_{34} \end{array}$$(56)$$\begin{array}{cc} {\dot{x}}_{35} & =k_{\mathrm{ISG}}x_{33}-d_{\mathrm{SAMHD}1}x_{35} \end{array}$$(57)$$\begin{array}{cc} {\dot{x}}_{36} & =k_{\mathrm{ISG}}x_{33}-f_{\mathrm{Vpu}}x_{24}x_{36}-d_{\mathrm{Teth}}x_{36}. \end{array}$$
It takes into account the counter-action of viral proteins Vif and Vpu. Here, we specify the parameter values as follows: $k_{\mathrm{STAT}}=0.1$ $h^{-1}$, $k_{\mathrm{ISG}}=2.94$ $h^{-1}$, $d_{\mathrm{IFN}}=0.15$ $h^{-1}$, $d_{\mathrm{STAT}}=0.03$ $h^{-1}$, $d_{\mathrm{APO}}=0.087$ $h^{-1}$, $d_{\mathrm{SAMHD}1}=0.16$ $h^{-1}$, $d_{\mathrm{Teth}}=0.044$ $h^{-1}$, $f_{\mathrm{Vif}}=f_{\mathrm{Vpu}}=7\times 10^{-6}$ $h^{-1}$.

### 2.3. Initial Value Problem

Parameters of the above set of reaction steps have been calibrated in our previous studies [14,15,16]. To compute the dynamics of all model components for the co-infection of CD4+ T cell with HIV-1 and TIP-2, the system of ordinary differential equations (ODEs) is supplemented with initial data corresponding to some scenario relevant for HIV virology:Initial number of free virions: $\mathrm{MOI}\equiv \left[V_{\mathrm{free}}\right]\left(0\right)=x_{1}\left(0\right)$;Initial number of free TIP-2 particles: $\left[{\mathrm{TIP}}_{\mathrm{free}}\right]\left(0\right)=x_{1T}\left(0\right)$;Initial number of extracellular interferon molecules: $\left[{\mathrm{IFN}}_{e}\right]\left(0\right)\equiv x_{32}\left(0\right)$;All other components are set to have zero values.

The relationship between the concentration and the number of protein molecules [${\mathrm{IFN}}_{e}$], [${\mathrm{IFN}}_{i}$], IRF3, [NF-kB], and [STAT12] was discussed in our previous work [14]. Note that the set of differential Equations (1)–(57) is not stiff; therefore, the initial value problem can be integrated numerically using the explicit Runge–Kutta method (e.g., ode45 in MATLAB).

### 2.4. Stochastic Model

Analysis reveals that the abundance of some components during the infection process remains low (on the order of a few particles/molecules), as illustrated in Figure 4 in Section 3.1. This is especially noticeable for the first twelve components. Consequently, the continuous trajectories obtained in the framework of the deterministic model are apparently not accurate, as the number of virions, TIPs, and molecules can inherently take on integer values.

To account for this discreteness, we translate the deterministic ODE model into a stochastic description based on a Markov chain (MC). This approach allows us to model integer-valued variables, obtain probability distributions instead of mean-field estimates, and compute the probabilities of productive cell infection at low MOI and TIPs.

We employ an effective translation method based on the Gillespie approach [17,18,19], originally developed for chemical kinetics but widely applicable to biological systems like viral dynamics. This approach requires no additional parameterization beyond the rates used in the ODE system (1)–(57).

A Markov chain can be described as a set of elementary transitions (reactions) with propensity (rate) for every transition. The propensity $a_{m}$ of the *m*th transition is defined by the relation: $a_{m}dt$ is the probability of the *m*th transition occurring within an infinitesimal time interval dt. The Markov chain of the stochastic model corresponding to the ODEs (1)–(57) is presented in Table 1. It contains M=118 transitions. Symbols $x_{n}^{--}$ and $x_{n}^{++}$ denote, respectively, the decrease and increase in abundance of component $x_{n}$ by one particle during the *m*th transition.

A number of approaches have been proposed for the numerical implementation of the MC. Most of them are based on Monte Carlo methods implemented with the use of the random number generator. Among them, the most popular is Gillespie’s direct method [18], which is applied here.

Gillespie’s direct method is described by Algorithm 1 in the same form as it was described in [14]. Here, $X\left(t\right)={\left[x_{1}\left(t\right),x_{2}\left(t\right)\ldots ,x_{N}\left(t\right)\right]}^{T}$ is the state vector describing abundances of all components at time *t*. Its initial value, $X_{0}=X\left(0\right)$, is the same as in the deterministic problem, provided that all initial abundances are integers.

In the case at hand, we are dealing with N=52 components participating in M=118 individual transitions. If we also have to compute $x_{37}=\left[V_{\mathrm{new}}\right]$, $x_{37T}=\left[{\mathrm{TIP}}_{\mathrm{new}}\right]$, and $x_{37\mathrm{HT}}=\left[{\mathrm{HIV}-\mathrm{TIP}}_{\mathrm{new}}\right]$, there is no need to add more transitions. It is enough to modify transitions 86, 89, and 92 in Table 1:(58)$$\begin{array}{ccc} m & \mathrm{Transition} & \mathrm{Propensity}\\ 86 & x_{26}^{--},x_{27}^{++},x_{37}^{++} & k_{\mathrm{mat}}x_{26}\\ 89 & x_{26T}^{--},x_{27T}^{++},x_{37T}^{++} & k_{\mathrm{mat}}x_{26T}\\ 92 & x_{26\mathrm{HT}}^{--},x_{27\mathrm{HT}}^{++},x_{37\mathrm{HT}}^{++} & k_{\mathrm{mat}}x_{26\mathrm{HT}} \end{array}$$
Then we are dealing with N=55 components with the same number of transitions: M=118.
**Algorithm 1** Gillespie’s direct methodInitialise the state vector $X⇐X_{0}$ and time t⇐0;**while**$t<t_{\mathrm{final}}$**do**    Compute propensities $a_{m}\left(X\right),m=1,\ldots ,M$;    Compute the cumulative sum $A_{m}=\sum_{i=1}^{m}a_{i},m=1,\ldots ,M$;    Generate two uniformly distributed random numbers $r_{1},r_{2}\in \left[0,1\right]$;    Compute the time interval to fire the next reaction $\Delta t=-\ln\left(r_{1}\right)/A_{M}$;    Determine the index *m* of the next reaction: find the smallest *m*: $A_{m}>r_{2}\cdot A_{M}$;    Update time t⇐t+Δt;    Update the state vector X in accordance with Table 1;    Save t,X;**end while**

Obtaining reliable statistics from the stochastic model requires a large number of simulations (typically $10^{5}$–$10^{6}$ realizations), which is computationally intensive for a large Markov chain. Furthermore, when component abundances become large, the time between stochastic events diminishes drastically, slowing down simulations. Since the dynamics of highly populated components are accurately described by deterministic ODEs, we implemented a hybrid algorithm to improve computational efficiency [14,16].

In the present work, we apply the same hybrid algorithm as described in [14]. In this algorithm, the dynamics for any component, $x_{n}$, are automatically switched from stochastic to deterministic once its abundance exceeds a predefined threshold $\bar{X}$. If the abundance later falls below $\bar{X}$, the dynamics are switched back to stochastic, with rounding the population to the nearest integer value. Thus, the stochastic and deterministic processes for different components are performed in parallel. We set a threshold of $\bar{X}=10^{4}$ for all components; only a few protein species reached this level. Comparisons of statistical characteristics (mean, median, etc.) computed by the fully stochastic model and the hybrid model with this threshold show negligible discrepancy, whereas the use of the hybrid scheme reduces computation time by a factor of four.

To further accelerate computations, we have implemented several optimizations:A binary search algorithm is employed to efficiently select the next reaction index *m*;Precomputed arrays included in the code are used to identify which components and propensities need updating after each transition, avoiding unnecessary recalculations;The algorithm was implemented in C++, utilizing arrays of pointers to functions. This allows the code to call only the specific propensity functions (for stochastic processes) and ODE right-hand sides (for deterministic processes) that are affected by the most recent transition, significantly reducing computational overhead [14].

All the above-mentioned improvements are very important when the number of components and transitions is high, as in the model studied here. The computations were run on an Intel Xeon E3-1220 v5 CPU 3GHz×4.

## 3. Results

### 3.1. Deterministic Model: Co-Infection Kinetics

The dynamics of all components of the HIV-1 and TIP-2 life cycle, as described by the deterministic model, are shown in Figure 4. The following set of scenarios is examined: HIV-1 MOI=10, $\left[{\mathrm{IFN}}_{e}\right]\left(0\right)=5$ molecules, and several numbers of defective particles TIP-2: 0,1,2,4,6,8,10.

One can see the components of the HIV-1 life cycle that are affected by the presence of TIP-2 particles. The co-infection of the cell with TIP-2 has a strong effect on the production of infectious HIV-1. Even one TIP-2 coinfecting the cell with ten HIV-1 reduces the number of released HIV-1 virions by about 30%. The competitive effect of TIP-2 on HIV-1 protein production starts with the $\left[{\mathrm{Gag}-\mathrm{Pol}}_{\mathrm{mem}}\right]$ protein, i.e., co-infection with only one TIP-2 dramatically reduces its abundance. Note that co-infection with TIP-2 increases the abundance of proteins related to the IFN-I system activation and the IFN-I production. An increase in the number of TIP-2 particles entering the cell results in a monotone decrease in the net virus secretion, an increase in the net TIP-2 production, and a bell-shaped dependence of the generated heterozygous HIV-1-TIP-2 particles.

### 3.2. Net Effect of Variation in TIP-2 and MOI on Life Cycle Efficacy

To evaluate the efficacy of the HIV-1 life cycle in the presence of TIP-2 particles, we considered the following equations. The cumulative number of new virions released by time *t*$\left[V_{\mathrm{new}}\right]\left(t\right)\equiv x_{37}\left(t\right)$ is described by the following equations:(59)$$\frac{d}{dt}\left[V_{\mathrm{new}}\right]=k_{\mathrm{mat}}\left[V_{\mathrm{bud}}\right]\Leftrightarrow {\dot{x}}_{37}=k_{\mathrm{mat}}x_{26}.$$
Similarly, the total numbers of released TIP-2 and heterozygous particles by infected CD4+ T cells by time *t*, $\left[{\mathrm{TIP}}_{\mathrm{new}}\right]\left(t\right)\equiv x_{37T}\left(t\right)$, $\left[{\mathrm{HIV}-\mathrm{TIP}}_{\mathrm{new}}\right]\left(t\right)\equiv x_{37\mathrm{HT}}\left(t\right)$, follow(60)$$\begin{array}{cc} \frac{d}{dt}\left[{\mathrm{TIP}}_{\mathrm{new}}\right] & =k_{\mathrm{mat}}\left[{\mathrm{TIP}}_{\mathrm{bud}}\right]\Leftrightarrow {\dot{x}}_{37T}=k_{\mathrm{mat}}x_{26T} \end{array}$$(61)$$\begin{array}{cc} \frac{d}{dt}\left[{\mathrm{HIV}-\mathrm{TIP}}_{\mathrm{new}}\right] & =k_{\mathrm{mat}}\left[{\mathrm{HIV}-\mathrm{TIP}}_{\mathrm{bud}}\right]\Leftrightarrow {\dot{x}}_{37\mathrm{HT}}=k_{\mathrm{mat}}x_{26\mathrm{HT}}. \end{array}$$
The kinetics of secretion of viral and defective particles for the same initial data as in Figure 4 are shown in Figure 5. An increase in the number of TIP-2 entering the cell from 1 to 10 reduces the net secretion of infectious virions from about 110 to 50 (see Figure 5(left)), while the number of released TIP-2 increases from 60 to about 105 TIPs (Figure 5(middle)). The production of heterozygous HIV-1-TIP-2 particles varies little between 55 and 40 (Figure 5(right)).

As for the total amount of secreted virions, TIP-2, and heterozygous particles produced by infected cells before they die, we consider the following metrics:secreted infectious HIV-1:(62)$$\left[V_{\mathrm{tot}}\right]=\left[V_{\mathrm{new}}\right]\left(T\right)=\int_{0}^{T}k_{\mathrm{mat}}\left[V_{\mathrm{bud}}\right]\;dt;$$secreted TIP-2 particles:(63)$$\left[{\mathrm{TIP}}_{\mathrm{tot}}\right]=\left[{\mathrm{TIP}}_{\mathrm{new}}\right]\left(T\right)=\int_{0}^{T}k_{\mathrm{mat}}\left[{\mathrm{TIP}}_{\mathrm{bud}}\right]\;dt;$$secreted HIV-1-TIP-2 particles:(64)$$\left[{\mathrm{TIP}}_{\mathrm{tot}}\right]=\left[{\mathrm{TIP}}_{\mathrm{new}}\right]\left(T\right)=\int_{0}^{T}k_{\mathrm{mat}}\left[{\mathrm{HIV}-\mathrm{TIP}}_{\mathrm{bud}}\right]\;dt.$$

For the duration of the viral replication cycle, assumed to be about T=36 h (the lifetime of a productively infected CD4+ T cell [20]), the following global characteristics of co-infection were computed for various combinations of MOI and number of TIP-2 entering the cell:(65)$$\left[V_{\mathrm{tot}}\right]\approx \left[V_{\mathrm{new}}\right]\left(36\;h\right),\;\left[{\mathrm{TIP}}_{\mathrm{tot}}\right]\approx \left[{\mathrm{TIP}}_{\mathrm{new}}\right]\left(36\;h\right),\;\left[{\mathrm{HIV}-\mathrm{TIP}}_{\mathrm{tot}}\right]\approx \left[{\mathrm{HIV}-\mathrm{TIP}}_{\mathrm{new}}\right]\left(36\;h\right).$$
The results are summarized in Figure 6 for scenarios reflecting the availability of extracellular IFN-I per cell at the level of $\left[{\mathrm{IFN}}_{e}\right]\left(0\right)=5$ molecules.

The curves shown in Figure 6 suggest that the reduction of the total number of HIV-1 secreted follows an exponential type of dependence on the co-infecting TIP-2 number. However, it does not result in a complete block of the virion production. Indeed, for MOI=1 and co-infecting ten TIP-2, the $[V_{\mathrm{tot}}]=1.5474$. The induction of intracellular defense systems via the IFN-I-induced signaling cascade, leading to activation of antiviral genes (ISGs) in parallel to co-infection, results in the blockade of mature HIV-1 production. The model predicts that the availability of five or ten extracellular IFN-I molecules, inducing the ISG-regulated antiviral proteins, reduces the amount of produced HIV-1 below one (0.2 and 0.09, respectively). Figure 6 shows that increasing the initial TIP-2 number results in the growth in the number of produced homozygous and heterozygous TIP-2 particles, which saturates at some levels, being higher for larger MOI, with an upper limit of about 100 particles.

### 3.3. Stochastic Model

The abundance of many components of the HIV-1 replication cycle is low. Hence, the randomness of the individual reaction steps has to be considered for a realistic quantification of the effect of TIP-2 on HIV replication. To this end, we employ the stochastic version of the HIV-1 and TIP-2 life cycle model formulated in Section 2. An example of an ensemble of realizations generated by the stochastic model is shown in Figure 7. The line colors are selected such that the trajectories of realizations with a larger number of released HIV-1 virions, [$V_{\mathrm{tot}}$], are closer to the red end of the spectrum, whereas those with a smaller number of released virions are closer to the blue end. The deterministic trajectory is shown by smooth black curves. The difference between the kinetics of co-infection characteristics described by the deterministic models and the stochastic version is noticeable for the populations of secreted virions and particles. Indeed, the fluctuations in [pre-V], [pre-TIP], [$V_{\mathrm{bud}}$], [${\mathrm{TIP}}_{\mathrm{bud}}$], and [${\mathrm{HIV-TIP}}_{\mathrm{bud}}$] are substantial.

### 3.4. Properties of Stochastic Ensemble Realizations

Infection of cells with a number of HIV-1 virions and TIP-2 particles results in multiple proviral and TIP-2 DNA copies integrated into the chromosome. When the randomness in the life-cycle reactions is taken into account, this results in a set of stochastic ensemble realizations. Probability density functions (pdf) allow one to assess how the random variables are distributed over the range of possible values. We used the stochastic model to characterize the patterns and sample statistics of the components of the HIV-1 and TIP-2 life cycles.

We use the stochastic version to evaluate the variation in the kinetics of production of HIV-1, TIP-2, and HIV-1-TIP-2 particles ($\left[V_{\mathrm{new}}\right]$, $\left[{\mathrm{TIP}}_{\mathrm{new}}\right]$, $\left[{\mathrm{HIV}-\mathrm{TIP}}_{\mathrm{tot}}\right]$). The results are shown in Figure 8. The variation in the final number of secreted virions, TIP-2, and mixed particles is about 10-fold.

The evolution of the probability density functions for all components of the HIV-1 and TIP-2 life cycle for various initial values of MOI and TIP-2 co-infection is presented in Figure 9 for the initial values: MOI=10, $\left[{\mathrm{TIP}}_{\mathrm{free}}\right]\left(0\right)=6$, $[{\mathrm{IFN}}_{e}]=5$. Here, a darker color corresponds to a higher value of the PDF related to the histogram. The evolution of the mean value is plotted by a blue curve; the evolution of the median is marked by the magenta line. The deterministic model trajectory is plotted by a green line.

The kinetics of the co-infection components, which are small in number, show a stairway character of the respective median lines. It is noticeable for the first twelve components and also for [pre-Virion], [pre-TIP], [$V_{\mathrm{bud}}$], [${\mathrm{TIP}}_{\mathrm{bud}}$], and [RIG1]. The other components show a smoother pattern, but the staircase behavior could still be noticed if zoomed in.

The histograms in Figure 9 display a multi-humped pattern for a multiplicity of co-infections larger than one. Each peak of the histograms corresponds to discrete numbers of integrated viral/TIP genomes observed in real HIV infection [21]. It is noticeable for components from [mRNAg] to [Vif] and also for [${\mathrm{RNA}}_{\mathrm{mem}}$], [${\mathrm{RNA}}_{\mathrm{mem}}^{\mathrm{TIP}}$], and [${\mathrm{Vpu}}_{\mathrm{mem}}$]. Simulations for other MOI and TIP-2 numbers show that the larger values of MOI and [TIP](0) increase the number of local maxima in the pdfs. The median value curves follow these maxima in the histograms, displaying a quasi-stair-like pattern with a smooth transition from lower to higher maximum, as is seen for [${\mathrm{mRNAc}}_{g}$], [Gag-Pol], [Gag], etc.

However, the structure of multi-humped behaviors of pdf disappears in histogram for the final products of the co-infection, i.e., [pre-Virion], [pre-TIP], [pre-HIV-TIP], [$V_{\mathrm{bud}}$], [${\mathrm{TIP}}_{\mathrm{bud}}$], [${\mathrm{HIV-TIP}}_{\mathrm{bud}}$], [$V_{\mathrm{mat}}$], [${\mathrm{TIP}}_{\mathrm{mat}}$], [${\mathrm{HIV-TIP}}_{\mathrm{mat}}$] and for the components of the IFN-I response elements (see plots from [RIG1] to [Tetherin]).

Quantitative analysis of the sample histograms suggest that they are not Gaussian. Hence, to characterize the sample statistics, we have evaluated the median and mean values, and the confidence intervals. The time evolution of the confidence intervals for [$V_{\mathrm{new}}$](t), [${\mathrm{TIP}}_{\mathrm{new}}$](t) and [${\mathrm{HIV-TIP}}_{\mathrm{new}}$](t) is shown in Figure 10 for the initial conditions MOI = 10, [${\mathrm{TIP}}_{\mathrm{free}}]\left(0\right)=6$, [${\mathrm{IFN}}_{e}]\left(0\right)=5$. The colored patches in Figure 10 provide quantitative details of the evolution of the histograms. The 25–75% confidence intervals (which include 50% of all realisations) are marked by yellow patches. The 15–85% confidence intervals are shown by the light-blue patches. They overlap with the 25–75% confidence intervals. The widest 5–95% confidence intervals (which include 90% of all realisations) are shown by the light-pink patches. There is a substantial difference between the mean, median curves and the deterministic solution. The confidence intervals turn out to be rather wide. For example, the lower boundary of the 25–75% confidence interval reaches a zero level. The median and mean curves go mostly above the deterministic solution, which indicates that the deterministic model underestimates productive co-infection of CD4+ T cell.

Notice that the mean value trajectory is located above both median and the deterministic curves. It has the following implication. When the population dynamics of the infection of CD4 T-cells is considered then the force of infection from a deterministic view depends on the mean number of [$V_{\mathrm{tot}}$] and the total number of infected T-cells. Therefore, the prediction that the mean value of [$V_{\mathrm{tot}}$] significantly exceeds the number of $V_{\mathrm{tot}}$ obtained in the framework of deterministic model (as seen in Figure 10(left)) emphasizes the relevance of the stochastic approach to properly understand the quantitative dynamics of infection spreading.

### 3.5. Impact of MOI and TIP-2 on Net HIV-1, TIP-2 and Heterozygous Particles Release

The stochastic model is used to quantify the impact of MOI and TIP-2 on the net HIV-1, TIP-2 and heterozygous particles release by co-infected CD4+ T cell during 36 h post infection. The dependence of the mean values of secreted infectious HIV-1 [$V_{\mathrm{tot}}$] on MOI and initial number of TIP-2 particles, [${\mathrm{TIP}}_{\mathrm{free}}$](0), are shown in Figure 11 and Figure 12, respectively. Three scenarios of availability of extracellular IFN-I per cell are examined: $\left[{\mathrm{IFN}}_{e}\right]\left(0\right)=0,5,10$.

Figure 11 shows that the co-infection with TIP-2 reduces the production of mature virions. The dependence of the mean( [$V_{\mathrm{tot}}$]) on con-infecting TIP-2 is nonlinear (Figure 11) but it is very close to linear in relation to MOI (Figure 12). The reduction degree is rather modest, e.g., for MOI = 10 and ten TIP-2 entering the cell, the net amount of HIV secreted is about three-times lower than in the case of mono-infection. The reduction effect turns out to be smaller for large numbers of extracellular IFN-I molecules $\left[{\mathrm{IFN}}_{e}\right]$ activating the IFN-induced intracellular protection system.

In parallel, we explore the dependence of the mean numbers of homozygous and heterozygous TIP-2 particles, i.e., [${\mathrm{TIP}}_{\mathrm{tot}}$] and [${\mathrm{HIV-TIP}}_{\mathrm{tot}}$], on the initial number of co-infecting TIP-2 for various MOI. The results are shown in Figure 13 and Figure 14, respectively. The total amount of [${\mathrm{TIP}}_{\mathrm{tot}}$] secreted by the co-infected cell increases with increased MOI. However, the extent of increase is only 4-fold when TIP-2 number ranges from 1 to 10 (see Figure 13). Activation of the IFN-mediated protection, linked to an increase of the availability of extracellular IFN-I per cell from $\left[{\mathrm{IFN}}_{e}\right]\left(0\right)=0$ (left) to $\left[{\mathrm{IFN}}_{e}\right]\left(0\right)=5$ (middle) and finally to $\left[{\mathrm{IFN}}_{e}\right]\left(0\right)=10$ (right), reduces the enhancement effect. The effect of increase saturates at high number of co-infecting TIP-2.

The dependence of mean heterozygous particles ([${\mathrm{HIV-TIP}}_{\mathrm{tot}}$]) on [TIP](0), summarized in Figure 14, shows a non-monotonic behaviour in contrast to the deterministic model-based prediction shown in Figure 6. The predictions of the deterministic model and its stochastic counterpart for the net production of [$V_{\mathrm{tot}}$], [${\mathrm{TIP}}_{\mathrm{tot}}$] and [${\mathrm{HIV-TIP}}_{\mathrm{tot}}$] can be compared by examining the Figure 6 versus the middle plots in Figure 11, Figure 13 and Figure 14. The numbers, computed for initial conditions: MOI =10, [${\mathrm{TIP}}_{\mathrm{free}}]\left(0\right)=6$, [${\mathrm{IFN}}_{e}]\left(0\right)=5$, indicate that the deterministic model underestimates the production of homozygous virions and TIP-2 particles by approximately a factor of two, but over-predicts the amount of secreted heterozygous particles; specifically, the maximal output of [$V_{\mathrm{new}}$] is around 300 in Figure 6 against the maximum mean value of 730 in Figure 11.

### 3.6. Reproduction Factor for TIP-2 and HIV-1-TIP-2 Particles

To characterize the efficacy of homozygous and heterozygous TIP-2 production by the co-infected CD4+ T cell, we examine the dependence of the amplification factor for the ratio of the initial number of TIP-2 to HIV-1 infecting the cell. The model-predicted ratios of mean total numbers of released TIP-2 particles and mature virions, i.e., $\mathrm{mean}(\left[{\mathrm{TIP}}_{\mathrm{tot}}\right])$/ $\mathrm{mean}(\left[V_{\mathrm{tot}}\right])$, and their dependance on the initial ratio of TIP-2 and HIV-1 virions, [${\mathrm{TIP}}_{\mathrm{free}}$](0)/MOI, are presented in Figure 15. The MOI ranges from 1 to 10 virions per cell and three different values of extracellular interferon and three scenarios of availability of extracellular IFN-I per cell are examined.

One can see that the relative amplification factor becomes greater than 1 for MOI >4. However, the stronger activation of IFN-I-mediated defence taking place for $\left[{IFN}_{e}\right]\left(0\right)=5$ (middle), $\left[{IFN}_{e}\right]\left(0\right)=10$ (right), increases the MOI threshold up to 7 and 10, respectively. The relative amount of TIP-2 with respect to HIV-1 infecting the cell becomes larger than 1 for MOI > 4, but a similar increase in the IFN-I response requires the MOI to be 10 and larger than 10 to keep the relative reproduction factor for TIP-2 above 1. A similar analysis of the dependence on TIP/MOI of the ratio of the mean total number of released heterozygous HIV-1-TIP-2 ($\mathrm{mean}(\left[{\mathrm{HIV}-\mathrm{TIP}}_{\mathrm{tot}}\right])$/ $\mathrm{mean}(\left[V_{\mathrm{tot}}\right])$), to the mean amount of secreted HIV-1 is shown in Figure 16.

The curves in Figure 15 show that dependence of $\mathrm{mean}(\left[{TIP}_{\mathrm{tot}}\right])$/$\mathrm{mean}(\left[V_{\mathrm{tot}}\right])$ versus $\left[{\mathrm{TIP}}_{\mathrm{free}}\right]\left(0\right)$/MOI is close to a linear one for the considered range of MOI. Hence, a slope of the curves could be used to characterize the relative reproduction factor of TIP-2/MOI during the co-infection cycle. Define this slope *S* as follows:(66)$$S=\frac{\mathrm{mean}\left(\left[{\mathrm{TIP}}_{\mathrm{tot}}\right]\right)/\mathrm{mean}\left(\left[V_{\mathrm{tot}}\right]\right)}{\left[{\mathrm{TIP}}_{\mathrm{free}}\right]\left(0\right)/\mathrm{MOI}}.$$
The values of the slope *S* for a range of MOI and available extracellular IFN-I molecules the estimated by least squares fitting are presented in Table 2.

Strong activation of the IFN-I-driven intracellular defence system reduces the enhancing effect of MOI on the production of defective TIP-2.

Overall, the model predicts that the competition between TIP-2 and HIV-1 depends on the multiplicity of infection and the activity of the IFN-I system. As the major therapeutic goal of TIP-2 application is to outcompete HIV-production, care should be given to quantitative levels of MOI and IFN-I in designing the protocols of TIP-2-based treatments.

### 3.7. TIP-2 Effect on HIV-1 Life Cycle Efficiency

The efficiency of the HIV-1 life cycle can be characterised by the ratio of the total viral progeny to the number of virions infecting the cell (MOI). This characteristic was defined as the Life Cycle Efficiency (LCE) in our previous study [14]:(67)$$\mathrm{Life}\;\mathrm{Cycle}\;\mathrm{Efficiency}=\frac{\mathrm{mean}(\left[V_{\mathrm{tot}}\right])}{\mathrm{MOI}}$$
The inhibitory effect of TIP-2 on HIV-1 LCE was estimated using the stochastic model and is shown in Figure 17.

The behaviour of the curves in Figure 17 suggest that LCE decreases with an increase of the number of co-infection TIP-2 particles. However, the degree of reduction is not strong and 10-fold increase of TIP-2 results in about three-times reduction of LCE. In addition, the extent of decrease in LCE lowers with the increase of [TIP](0). The plots also show that LCE only slightly depends on MOI. Indeed, there is a severe congestion of curves plotted for different MOI. Maximal values of LCE reached for [TIP](0)=0 are very similar for different MOI.

Another characteristic of the effect of TIP-2 on HIV-1 production, introduced for IFN-I-based inhibition in our previous work [14], can be used to quantify the suppression degree of viral production by defective viral particles, TIP-2. The Inhibitory Factor (IF) can be defined as the ratio of the total number of released virions in presence of [${\mathrm{TIP}}_{\mathrm{free}}$](0) to the case mono-infection, i.e., the absence of TIP-2:(68)$$\mathrm{TIP}-2\;\mathrm{Inhibitory}\;\mathrm{Factor}\;=\frac{\mathrm{mean}\left(\left[V_{\mathrm{tot}}\right]\right)\mathrm{at}\left[{\mathrm{TIP}}_{\mathrm{free}}\right]\left(0\right)>0}{\mathrm{mean}\left(\left[V_{\mathrm{tot}}\right]\right)\mathrm{at}\left[{\mathrm{TIP}}_{\mathrm{free}}\right]\left(0\right)=0}.$$
Note that using a stochastic ensemble realizations modelling, we compare the (sample) mean values for viral production. The model predictions of the dependence of the IF on MOI are summarized in Figure 18 for different values of [TIP](0) and for three initial amounts of extracellular IFN-I molecules: $\left[{\mathrm{IFN}}_{e}\right]\left(0\right)=0$ (left), $\left[{\mathrm{IFN}}_{e}\right]\left(0\right)=5$ (middle), $\left[{\mathrm{IFN}}_{e}\right]\left(0\right)=10$ (right).

The results shown in Figure 17 indicate that the Inhibitory Factor rapidly declines with the increase of co-infecting TIP-2, but slowly grows with the increase of MOI.

### 3.8. Extinction Probability for HIV-1 Life-Cycle

In the framework of stochastic model, the realisations with zero number of released mature virions, i.e., [$V_{\mathrm{tot}}$] = 0, represent degenerate or extinct infection cycles. The developed stochastic model enables computing the probability of extinct cases, $P_{e}$, in relation to the initial number of infecting virions and the amount of extracellular IFN-I molecules. The corresponding results for the probabilities of productive infection of a target cell, which is $1-P_{e}$, are shown in Figure 19 and Figure 20. In the absence of [${\mathrm{IFN}}_{e}$], the probability of productive infection $1-P_{e}$ in Figure 19(left) is close to one for MOI > 7. The probability of abortive life cycle is shown in Figure 19(right) in a logarithmic scale.

When extracellular interferon is available, i.e., $\left[{\mathrm{IFN}}_{e}\right]\left(0\right)=5$ and 10, the probability of productive infection is almost independent of TIP-2, but strongly reduces for lower MOI. It is close to 0.8 for MOI = 10 $\left[{\mathrm{IFN}}_{e}\right]\left(0\right)=5$ but goes down to 0.6 $\left[{\mathrm{IFN}}_{e}\right]\left(0\right)=10$.

## 4. Discussion

In our study, we have have developed a mathematical model describing the impact of therapeutic interfering viral particles (TIP-2) on the HIV-1 life cycle of in CD4+ T cells. The model captures interactions between two parallel sets of biochemical reactions underlying HIV-1 and TIP-2 replication, resulting in production of (i) homozygous mature virions, (ii) TIP-2 particles, and (iii) heterozygous HIV-1-TIP-2 particles. Our work is based on our previously formulated theoretical model of the HIV-1 life cycle with the interferon type I (IFN-I) response [14]. The molecular-biological characteristics of TIP-2 particles required for the calibration of the respective set of equations are presented in a recent study by Pitchai et al. [8]. The inclusion of heterozygous RNA genomes is supported by experimental data demonstrating that RNA packaging during particle assembly is an efficient process leading to the generation of heterozygous particles [22]. Using deterministic and stochastic descriptions of the intracellular dynamics of HIV-1 and TIP-2 co-infection with and the antiviral type I IFN response, we examined the efficacy of TIP-2 in reducing the infectious virions production for a broad range of MOI, initial number of TIP-2 and the amount of extracellular IFN-I molecules activating ISGs.

Type I interferon proteins are an important component of cellular antiviral defence mechanisms. They are activated after virus infections through specialized pattern recognition receptors that recognize viral RNAs or DNAs in different locations in the cell. The produced IFNs then activate the expression of IFN-stimulated genes in infected cells and in surrounding cells and trigger an antiviral state that reduces virus production. Infections with HIV-1 are no exceptions, and IFNs are produced throughout the entire infection course [23]. The TIPs described by Pitchai et al. [8] and studied here contain structural HIV-1 elements like the TAR stem-loop structure and the U-rich region of the LTR in their genomes, which are key inducers of the IFN-I response [24]. Furthermore, in TIP-treated animals ([8] Supplementary Materials, Figure S14), qRT-PCR analysis of RNAs from retropharyngeal lymph node tissue revealed no significant differences between TIP-treated and control animals in the expression of selected inflammatory cytokines, including IFN-alpha and IFN-beta. Thus, the assumption used in constructing our model that TIPs induce an IFN-I response similar to that of wild-type HIV-1 appears justified.

Our theoretical study aims to reconcile results of the promising in vivo application of TIP-2 to reduce SHIV-induced disease in non-human primate [8] with the results of human studies showing the lack of beneficial effect of defective interfering particles (DIPs) in people living with HIV-1 [9]. Our theoretical modelling suggests that co-infection with TIP-2 of productively infected CD4+ T cells leads to partial inhibition of produced infectious virions. However, even at a high TIP-2 to HIV-1 ratio of 10:1, the infected cell still produces some viruses although in a minor quantity. Notably, the activation of IFN-I-induced ISGs in parallel with the co-infection process completely blocks (deterministic model) or substantially reduces (stochastic model) the production of mature HIV-1 by the infected cell. It can be assumed that in order to increase the efficacy of reducing HIV production via TIPs, it is necessary to consider their combination with other types of antiviral defence mechanisms. Among these, the IFN system should be considered. However, the model predicts that the outcome of the competition between TIP-2 and HIV-1 highly depends on MOI and activity of the IFN-I system. Indeed, at lower MOI, the relative advantage of TIPs decreases, and the same effect takes place for higher IFN-I levels. These features suggest that TIP therapy could be less effective under ART-suppressed conditions when combined with IFN stimulation.

A critical consideration when using HIV-1/TIP co-infection is the possibility of genetic recombination of chimeric viral particles. Indeed, the recombination could lead to a safety risk as HIV could recombine with TIP genomes, potentially restoring lost functions or generating new variants. Our theoretical model allows one to predict the production of HIV-1/TIP-2 heterozygous particles, which are a prerequisite for recombination to take place. In conjunction with the high mutation rate, this could favour the emergence of escape from TIP-2 mediated interference [12].

Results of the simulation show that the Inhibitory Factor, defined as the ratio of the total number of released virions in presence of [TIP](0) to the case of mono-infection, i.e., in the absence of TIP-2, rapidly increases with the increase of co-infecting TIP-2 but slowly grows with the increase of MOI. Furthermore, the HIV-1 Life Cycle Efficiency (LCE) estimated by the ratio $\mathrm{mean}(\left[V_{\mathrm{tot}}\right])/\mathrm{MOI}$ decreases with an increase of the number of co-infecting TIP-2 particles. However, the degree of reduction is not strong and a 10-fold increase of TIP-2 number results in only about 3-fold reduction of LCE. In addition, the degree of LCE reduction decreases with increase of [TIP](0).

In the absence of extracellular IFN-I, the probability of productive infection approaches unity for MOI>7. When extracellular IFN-I is present ($\left[{\mathrm{IFN}}_{e}\right]\left(0\right)=5$ or 10), the probability becomes almost independent of TIP-2 but is significantly reduced at lower MOI, dropping to approximately 0.8 and 0.6 for MOI=10 at $\left[{\mathrm{IFN}}_{e}\right]\left(0\right)=5$ and 10, respectively.

The examined model is formulated to describe a single HIV-1 replication cycle in the presence of TIP-2. The effect of co-infection with TIP-2 on production of infectious HIV-1 in our study is characterized by two integrative parameters: the TIP-2 inhibitory factor and Life Cycle Efficiency, presented in Figure 17 and Figure 18. The model predicts that both characteristics depend on MOI and reduce their values with a decrease in the number of HIV-1 infecting the cell. Overall, it implies that the reduction in viral load in patients under HAART could be associated with a reduced effectiveness of TIP-2 over time of infection.

The presented model analyzes the HIV and TIP replication cycle at the individual cell level. However, the efficacy of the TIP-based therapy at the patient level depends on population dynamics, i.e., the spread of TIPs vs HIV across target cells. The extension of the model from a single cell to the tissue level will be a necessary step for translation of the model findings to clinically relevant predictions.

The predicted synergy between IFN-I and TIPs followed from our model could be a factor explaining the success of TIP-2-based therapy in primates. However, other species-specific factors such as SHIV vs HIV, acute vs chronic infection, and baseline immunity must also be taken into account. Addressing the above question theoretically will require a further extension of the single-cell model to a systemic level description of HIV and SHIV infections.

The presented mathematical model can be used as a rational design tool (i) to conduct in vitro experiments with TIPs as combination partners to suppress HIV-1 replication in various target cells, and (ii) to generate hypotheses on TIP-incorporated combination therapy modalities. Our modelling study shows that the TIP-based treatment can reduce mature HIV-1 production to different extents. The model can be used as a building block for developing detailed multi-scale models of HIV infection in humans, which will allow quantitative predictions for effect of the TIP-based therapy on viral load, genomic diversification of defective particles and chronic immune activation. Importantly, the use of TIPs as a novel therapeutic strategy will likely require combination with other immune defence components. As we show at the single-cell level, it is the synergy between TIP-2 and IFN-mediated protection that completely blocks/substantially reduces productive HIV-1 infection. Overall, our study contributes to the search for novel therapeutic strategies to achieve sustained, antiretroviral-free control of HIV infection, which is regarded as a major medical priority [25].

## Figures and Tables

**Figure 1 viruses-17-01378-f001:**
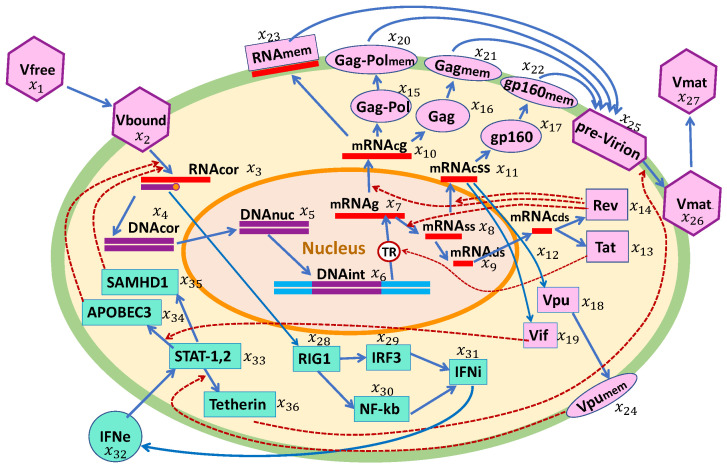
Biochemical scheme of the HIV-1 life cycle and the type I interferon (IFN-I) response. The following stages are considered: viral entry, reverse transcription, integration into the chromosome, transcription and splicing of viral RNAs, translation of proteins including the proteins inhibiting the action of ISGs, assembly of pre-virions, budding and release of immature virions, maturation of released virions, as well as the antiviral response induced by sensing of viral RNAs, IFN synthesis, and translation of antiviral proteins by IFN-stimulated genes.

**Figure 2 viruses-17-01378-f002:**
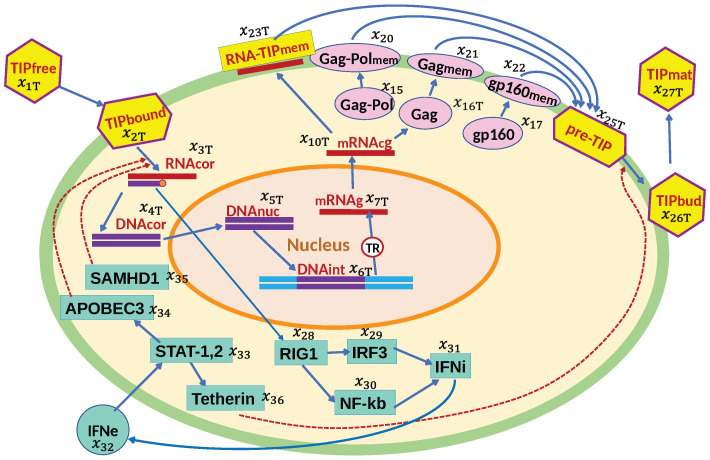
Biochemical scheme of the TIP-2 replication cycle. The consecutive chain of elementary processes comprises TIP entry, reverse transcription, integration into the chromosome, transcription and splicing of viral RNAs, assembly of pre-TIPs, budding and release of TIPs, sensing of TIP RNAs, IFN synthesis, and translation of antiviral proteins by IFN-stimulated genes.

**Figure 3 viruses-17-01378-f003:**
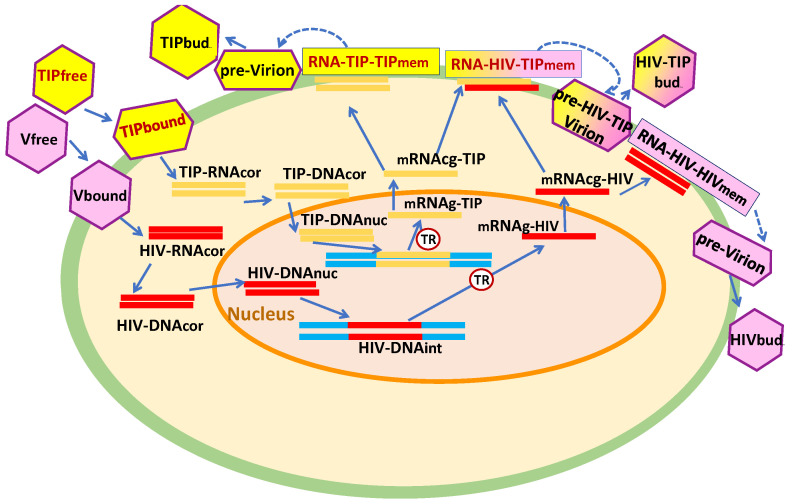
Biochemical scheme illustrating the formation of homozygous HIV-1 virions, homozygous TIP-2, and heterozygous HIV-TIP-2 particles.

**Figure 4 viruses-17-01378-f004:**
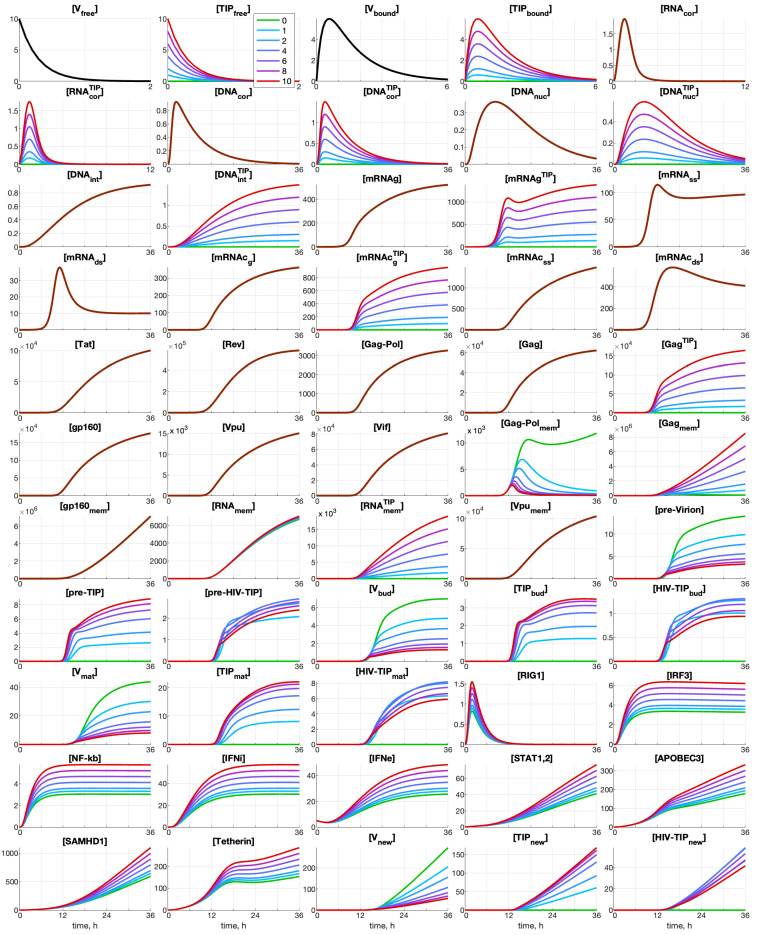
Solutions to the deterministic model of CD4+ T cell co-infection with HIV-1 and TIP-2 described by Equations (1)–(57). The initial values are MOI=10, $\left[{\mathrm{IFN}}_{e}\right]\left(0\right)=5$, $\left[{\mathrm{TIP}}_{\mathrm{free}}\right]\left(0\right)=0,1,2,4,6,8,10$. The line color corresponds to the number of initial TIP-2. They are defined in the plot for [${\mathrm{TIP}}_{\mathrm{free}}$]. The kinetics of released HIV-1 ([$V_{\mathrm{new}}$]), TIP-2 ([${\mathrm{TIP}}_{\mathrm{new}}$]), and HIV-1-TIP-2 ([${\mathrm{HIV-TIP}}_{\mathrm{new}}$]) particles are presented.

**Figure 5 viruses-17-01378-f005:**
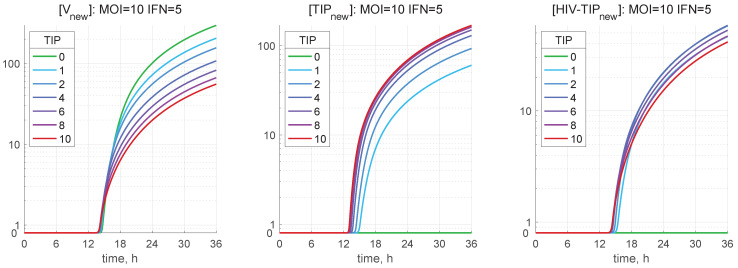
Dynamics of released mature virions (**left**), TIP-2 (**middle**), and HIV-1-TIP-2 particles (**right**) versus MOI=10, $\left[{\mathrm{IFN}}_{e}\right]\left(0\right)=5$, and different degree of co-infection [${\mathrm{TIP}}_{\mathrm{free}}$](0) (indicated in the legends).

**Figure 6 viruses-17-01378-f006:**
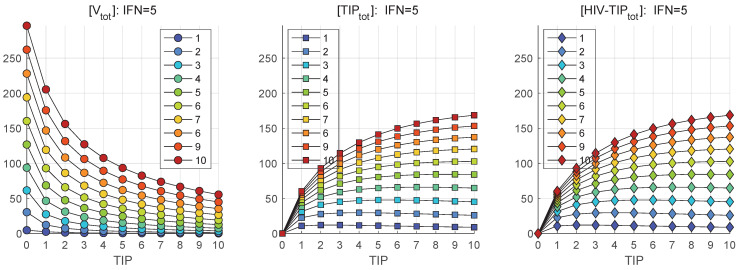
Total numbers of mature virions (**left**), TIP-2 particles (**center**), and mature heterozygous HIV-1-TIP-2 particles (**right**) produced over 36 h by the infected cell for different numbers of co-infecting TIP-2 $\left[{\mathrm{TIP}}_{\mathrm{free}}\right]\left(0\right)$ (denoted as TIP) and HIV-1 MOI (indicated in the legend) with an abundance of extracellular IFN-I per cell $\left[{\mathrm{IFN}}_{e}\right]\left(0\right)=5$ (denoted as IFN).

**Figure 7 viruses-17-01378-f007:**
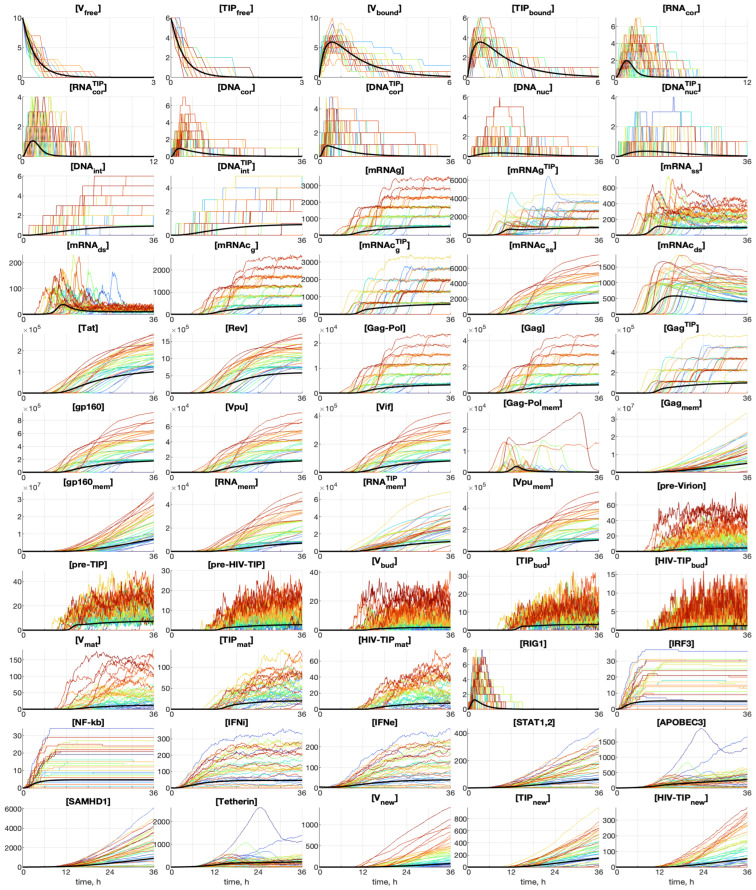
Ensemble of stochastic realizations for the co-infection scenario with the following initial data: MOI=100, $[{\mathrm{TIP}}_{\mathrm{free}}\left(0\right)]=100$, $\left[{\mathrm{IFN}}_{e}\right]\left(0\right)=5$. The line colors of the realizations with larger. numbers of released HIV-1 virions, [V_tot_], are closer to the red end of the spectrum whereas they are closer to the blue end for trajectories with smaller numbers of released virions. The deterministic solution is shown in black.

**Figure 8 viruses-17-01378-f008:**
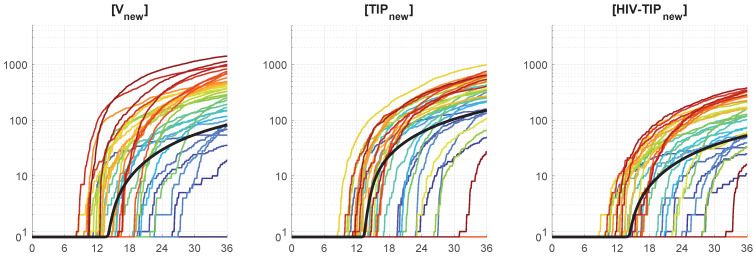
Examples of stochastic realizations for $\left[V_{\mathrm{new}}\right]$ (**left**), $\left[{\mathrm{TIP}}_{\mathrm{new}}\right]$ (**middle**), and $\left[{\mathrm{HIV}-\mathrm{TIP}}_{\mathrm{new}}\right]$ (**right**). The initial parameters are MOI=10, $\left[{\mathrm{TIP}}_{\mathrm{free}}\right]\left(0\right)=6$, $\left[{\mathrm{IFN}}_{e}\right]\left(0\right)=5$. The deterministic trajectory is plotted by a bold black line. These are the same plots as the last three depicted in Figure 7 but on a logarithmic scale.

**Figure 9 viruses-17-01378-f009:**
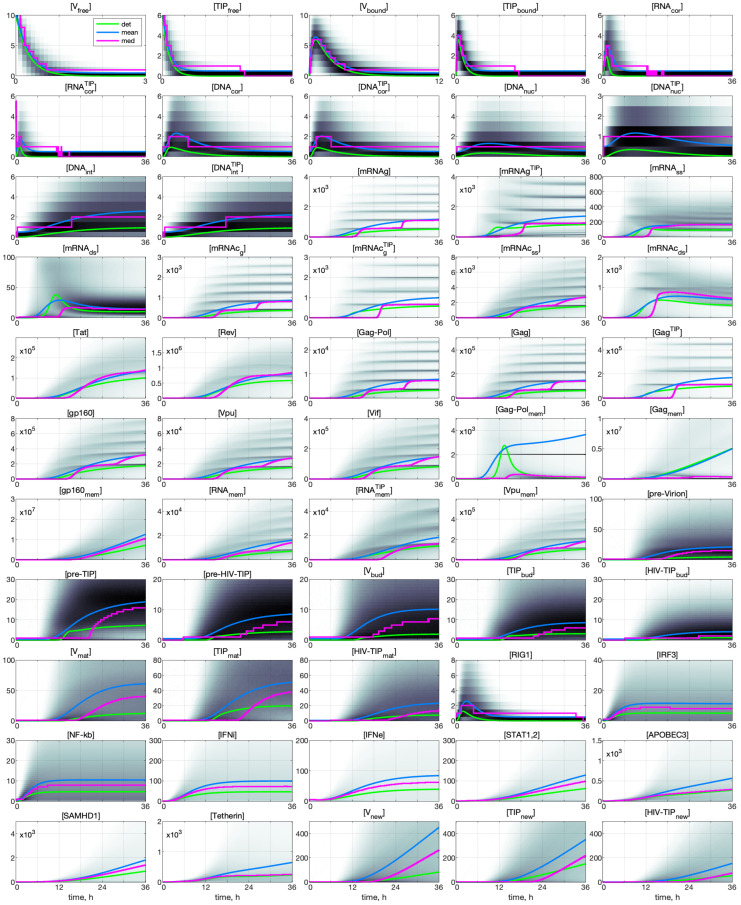
Examples of sliding histograms representing the evolution of the probability density function for initial values of MOI and TIP-2 are shown: MOI=10, $\left[{\mathrm{TIP}}_{\mathrm{free}}\right]\left(0\right)=6$, and $\left[{\mathrm{IFN}}_{e}\right]\left(0\right)=5$. color intensity scaling is adjusted to every component in order to outline local variations in the histogram. Darker color corresponds to a higher value of the PDF. The evolution of the mean values (mean), medians (med), and deterministic (det) trajectory are plotted by blue, magenta, and green lines, respectively.

**Figure 10 viruses-17-01378-f010:**
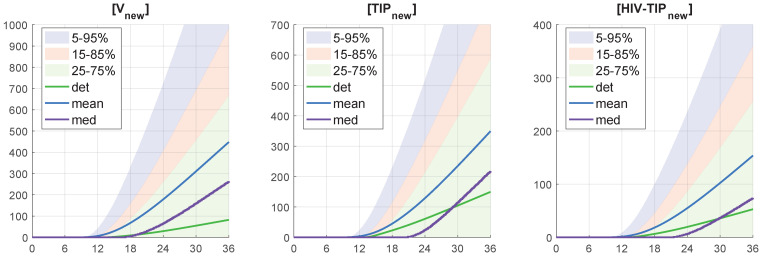
Time-varying confidence intervals for [$V_{\mathrm{new}}$], [${\mathrm{TIP}}_{\mathrm{new}}$] and [${\mathrm{HIV-TIP}}_{\mathrm{new}}$] computed for the initial values: MOI = 10, [${\mathrm{TIP}}_{\mathrm{free}}$](0) = 6 and [IFNe](0) = 5. The deterministic solution (det), mean values (mean), and medians (med) of the stochastic ensemble realizations are plotted (the color code for the patches and lines are explained in the legend).

**Figure 11 viruses-17-01378-f011:**
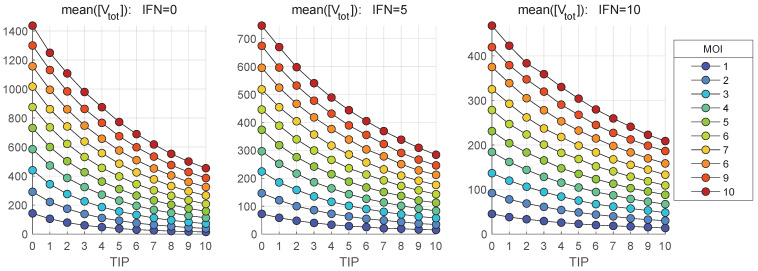
Mean value of the total number of mature virions, [$V_{\mathrm{tot}}$], released during the infection versus the initial number of co-infecting TIP-2 for different MOI (indicated in the legend). Three scenarios of availability of extracellular IFN-I per cell are examined: $\left[{\mathrm{IFN}}_{e}\right]\left(0\right)=0$ (**left**), $\left[{\mathrm{IFN}}_{e}\right]\left(0\right)=5$ (**middle**), $\left[{\mathrm{IFN}}_{e}\right]\left(0\right)=10$ (**right**).

**Figure 12 viruses-17-01378-f012:**
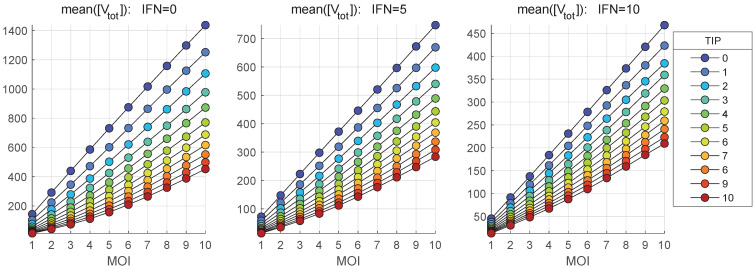
Mean value of the total number of mature virions, [$V_{\mathrm{tot}}$], released during the infection versus MOI for different number of co-infecting TIP-2 (indicated in the legend). Three scenarios of availability of extracellular IFN-I per cell are examined: $\left[{\mathrm{IFN}}_{e}\right]\left(0\right)=0$ (**left**), $\left[{\mathrm{IFN}}_{e}\right]\left(0\right)=5$ (**middle**), $\left[{\mathrm{IFN}}_{e}\right]\left(0\right)=10$ (**right**).

**Figure 13 viruses-17-01378-f013:**
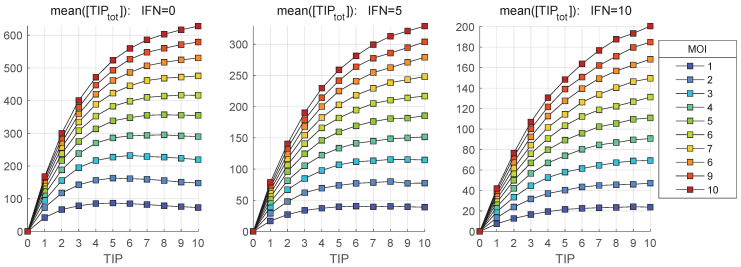
Mean value of the total number of mature TIP-2 particles, [${\mathrm{TIP}}_{\mathrm{tot}}$], released during the infection versus the initial number of coinfecting TIP-2 for different MOI (indicated in the legend). Three scenarios of availability of extracellular IFN-I per cell are examined: $\left[{\mathrm{IFN}}_{e}\right]\left(0\right)=0$ (**left**), $\left[{\mathrm{IFN}}_{e}\right]\left(0\right)=5$ (**middle**), $\left[{\mathrm{IFN}}_{e}\right]\left(0\right)=10$ (**right**).

**Figure 14 viruses-17-01378-f014:**
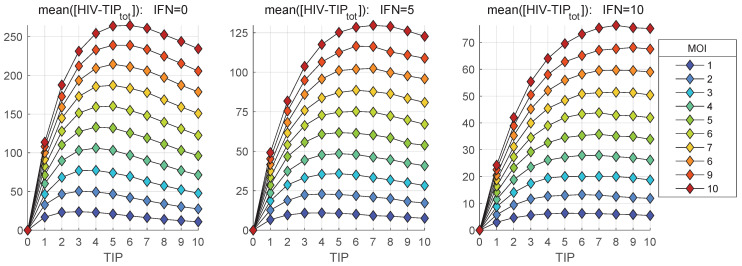
Mean value of the total number of mature heterozygous HIV-1-TIP-2 particles, [${\mathrm{HIV-TIP}}_{\mathrm{tot}}$], released during the infection versus the initial number of co-infecting TIP-2 for different MOI (indicated in the legend). Three scenarios of availability of extracellular IFN-I per cell are examined: $\left[{\mathrm{IFN}}_{e}\right]\left(0\right)=0$ (**left**), $\left[{\mathrm{IFN}}_{e}\right]\left(0\right)=5$ (**middle**), $\left[{\mathrm{IFN}}_{e}\right]\left(0\right)=10$ (**right**).

**Figure 15 viruses-17-01378-f015:**
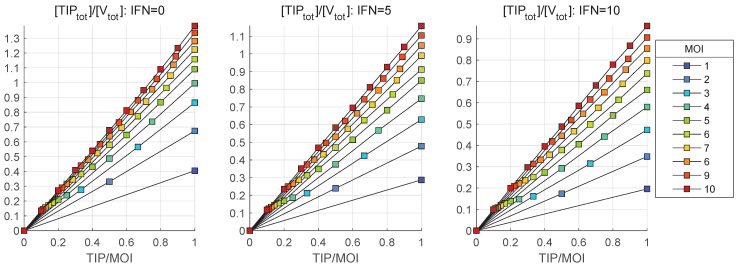
Ratio of mean values of the total produced TIP-2 particles to the total mature virions, i.e., the ratio $\mathrm{mean}\left(\left[{\mathrm{TIP}}_{\mathrm{tot}}\right]\right)/\mathrm{mean}\left(\left[V_{\mathrm{tot}}\right]\right)$ for different MOI (indicated in the legend) and for three initial amounts of extracellular IFN-I: $\left[{\mathrm{IFN}}_{e}\right]\left(0\right)=0$ (**left**), $\left[{\mathrm{IFN}}_{e}\right]\left(0\right)=5$ (**middle**) and $\left[{\mathrm{IFN}}_{e}\right]\left(0\right)=10$ (**right**).

**Figure 16 viruses-17-01378-f016:**
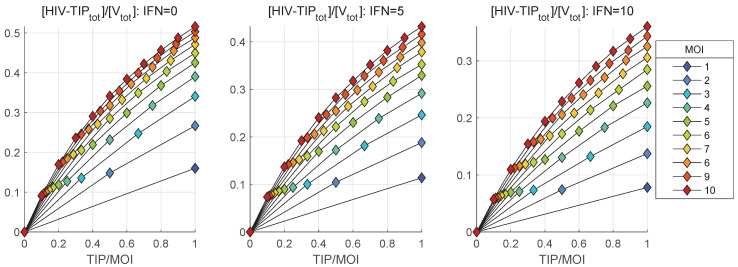
Ratio of the mean values of the total produced heterozygous HIV-1-TIP-2 particles to the total mature secreted HIV-1 virions, i.e., the value $\mathrm{mean}\left(\left[{\mathrm{HIV}-\mathrm{TIP}}_{\mathrm{tot}}\right]\right)/\mathrm{mean}\left(\left[V_{\mathrm{tot}}\right]\right)$, for different MOI (indicated in the legend) and for three initial amounts of the extracellular IFN: $\left[{\mathrm{IFN}}_{e}\right]\left(0\right)=0$ (**left**), $\left[{\mathrm{IFN}}_{e}\right]\left(0\right)=5$ (**middle**), $\left[{\mathrm{IFN}}_{e}\right]\left(0\right)=10$ (**right**).

**Figure 17 viruses-17-01378-f017:**
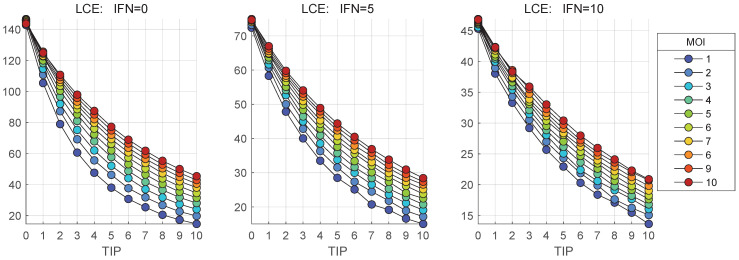
Life Cycle Efficiency (LCE) versus [TIP](0) for different MOI (indicated in the legend) and for three initial amounts of extracellular IFN-I molecules: $\left[{\mathrm{IFN}}_{e}\right]\left(0\right)=0$ (**left**), $\left[{\mathrm{IFN}}_{e}\right]\left(0\right)=5$ (**middle**), $\left[{\mathrm{IFN}}_{e}\right]\left(0\right)=10$ (**right**).

**Figure 18 viruses-17-01378-f018:**
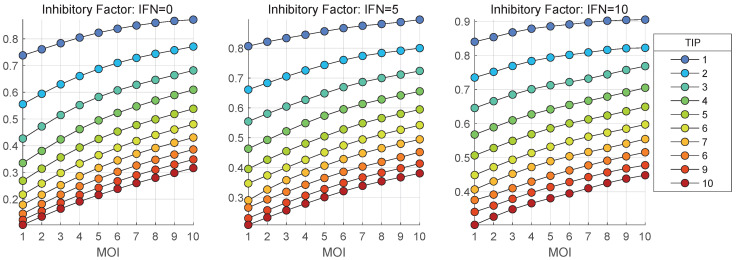
TIP-2 Inhibitory Factor defined by Equation (Equation 68) versus MOI for different numbers of TIP-2 particles [TIP](0) (indicated in the legend) and for three initial amounts of extracellular IFN-I molecules outside the cell: $\left[{\mathrm{IFN}}_{e}\right]\left(0\right)=0$ (**left**), $\left[{\mathrm{IFN}}_{e}\right]\left(0\right)=5$ (**middle**), $\left[{\mathrm{IFN}}_{e}\right]\left(0\right)=10$ (**right**).

**Figure 19 viruses-17-01378-f019:**
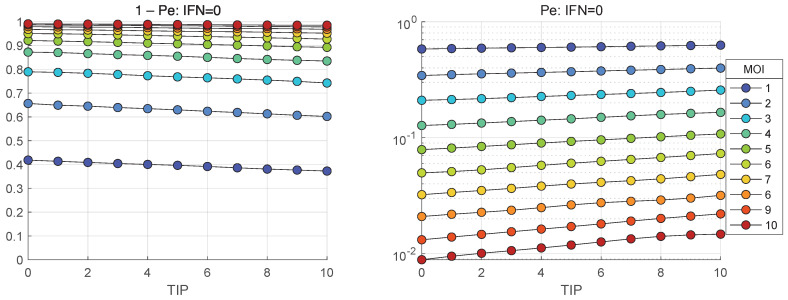
Probability of productive infection, $1-P_{e}$ (**left**). Probability of abortive infection $P_{e}$ (**right**). The dependence on initial number of coinfecting TIP-2 is shown for $\left[{\mathrm{IFN}}_{e}\right]\left(0\right)=0$.

**Figure 20 viruses-17-01378-f020:**
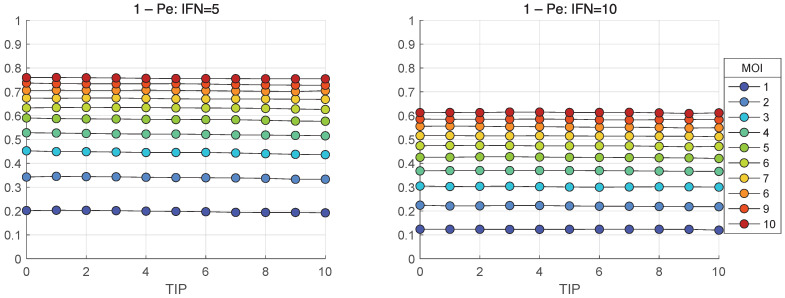
Probability of productive infection $1-P_{e}$ as function of TIP(0) and for various MOI when the extracellular IFN-I is available: $\left[{\mathrm{IFN}}_{e}\right]\left(0\right)=5$ (**left**) and for $\left[{\mathrm{IFN}}_{e}\right]\left(0\right)=10$ (**right**).

**Table 1 viruses-17-01378-t001:** The Markov chain elements of the model. Here, for brevity, the C++ notation is used: $x^{++}\Leftrightarrow x\to x+1$ and $x^{--}\Leftrightarrow x\to x-1$. For compactness we denote $\Theta_{c}=F_{c}\theta$, $\Theta_{c}^{T}=F_{c}^{\mathrm{TIP}}\theta_{T}$, $\Theta_{c}^{\mathrm{HT}}=F_{c}^{\mathrm{HT}}\theta_{\mathrm{HT}}$.

*m*	Transition	Propensity	*m*	Transition	Propensity	*m*	Transition	Propensity
1	$x_{1}^{--},x_{2}^{++}$	$k_{\mathrm{bound}}x_{1}$	41	$x_{10T}^{--},x_{23T}^{++}$	$k_{\mathrm{tp},\mathrm{RNA}}x_{10T}$	81	$x_{25T}^{--},x_{26T}^{++}$	$k_{\mathrm{bud}}x_{25T}$
2	$x_{1}^{--}$	$d_{\mathrm{free}}x_{1}$	42	$x_{10T}^{--}$	$d_{{\mathrm{RNA}}_{g}}x_{10T}$	82	$x_{25T}^{--}$	$d_{\mathrm{comb}}x_{25T}$
3	$x_{1T}^{--},x_{2T}^{++}$	$k_{\mathrm{bound}}x_{1T}$	43	$x_{11}^{--}$	$d_{{\mathrm{RNA}}_{\mathrm{ss}}}x_{11}$	83	$x_{25\mathrm{HT}}^{++}$	$k_{\mathrm{comb}}\Theta_{c}^{\mathrm{HT}}$
4	$x_{1T}^{--}$	$d_{\mathrm{free}}x_{1T}$	44	$x_{12}^{--}$	$d_{{\mathrm{RNA}}_{\mathrm{ds}}}x_{12}$	84	$x_{25\mathrm{HT}}^{--},x_{26\mathrm{HT}}^{++}$	$k_{\mathrm{bud}}x_{25\mathrm{HT}}$
5	$x_{2}^{--},x_{3}^{++}$	$k_{\mathrm{fuse}}x_{2}$	45	$x_{13}^{++}$	$k_{\mathrm{trans}}f_{\mathrm{ds},\mathrm{Tat}}x_{12}$	85	$x_{25\mathrm{HT}}^{--}$	$d_{\mathrm{comb}}x_{25\mathrm{HT}}$
6	$x_{2}^{--}$	$d_{\mathrm{bound}}x_{2}$	46	$x_{13}^{--}$	$d_{\mathrm{Tat}}x_{13}$	86	$x_{26}^{--},x_{27}^{++}$	$k_{\mathrm{mat}}x_{26}$
7	$x_{2T}^{--},x_{3T}^{++}$	$k_{\mathrm{fuse}}x_{2T}$	47	$x_{14}^{++}$	$k_{\mathrm{trans}}f_{\mathrm{ds},\mathrm{Rev}}x_{12}$	87	$x_{26}^{--}$	$f_{\mathrm{Teth}}x_{36}x_{26}$
8	$x_{2T}^{--}$	$d_{\mathrm{bound}}x_{2T}$	48	$x_{14}^{--}$	$d_{\mathrm{Tat}}x_{14}$	88	$x_{26}^{--}$	$d_{\mathrm{bud}}x_{26}$
9	$x_{3}^{--},x_{4}^{++}$	$k_{\mathrm{RT}}x_{3}$	49	$x_{15}^{++}$	$k_{\mathrm{trans}}f_{g,\mathrm{Gag}-\mathrm{Pol}}x_{10}$	89	$x_{26T}^{--},x_{27T}^{++}$	$k_{\mathrm{mat}}x_{26T}$
10	$x_{3}^{--}$	$d_{{\mathrm{RNA}}_{\mathrm{cor}}}x_{3}$	50	$x_{15}^{--},x_{20}^{++}$	$k_{\mathrm{tp},\mathrm{Gag}-\mathrm{Pol}}x_{15}$	90	$x_{26T}^{--}$	$f_{\mathrm{Teth}}x_{36}x_{26T}$
11	$x_{3}^{--}$	$f_{\mathrm{APO}}x_{34}x_{3}$	51	$x_{15}^{--}$	$d_{\mathrm{Gag}-\mathrm{Pol}}x_{15}$	91	$x_{26T}^{--}$	$d_{\mathrm{bud}}x_{26T}$
12	$x_{3}^{--}$	$f_{\mathrm{SAM}}x_{35}x_{3}$	52	$x_{16}^{++}$	$k_{\mathrm{trans}}f_{g,\mathrm{Gag}}x_{10}$	92	$x_{26\mathrm{HT}}^{--},x_{27\mathrm{HT}}^{++}$	$k_{\mathrm{mat}}x_{26\mathrm{HT}}$
13	$x_{3T}^{--},x_{4T}^{++}$	$k_{\mathrm{RT}}x_{3T}$	53	$x_{16}^{--},x_{21}^{++}$	$k_{\mathrm{tp},\mathrm{Gag}}x_{16}$	93	$x_{26\mathrm{HT}}^{--}$	$f_{\mathrm{Teth}}x_{36}x_{26\mathrm{HT}}$
14	$x_{3T}^{--}$	$d_{{\mathrm{RNA}}_{\mathrm{cor}}}x_{3T}$	54	$x_{16}^{--}$	$d_{\mathrm{Gag}}x_{16}$	94	$x_{26\mathrm{HT}}^{--}$	$d_{\mathrm{bud}}x_{26\mathrm{HT}}$
15	$x_{3T}^{--}$	$f_{\mathrm{APO}}x_{34}x_{3T}$	55	$x_{16}^{++}$	$k_{\mathrm{trans}}f_{g,\mathrm{Gag}}x_{10}$	95	$x_{27}^{--}$	$d_{\mathrm{free}}x_{27}$
16	$x_{3T}^{--}$	$f_{\mathrm{SAM}}x_{35}x_{3T}$	56	$x_{16T}^{--},x_{21}^{++}$	$k_{\mathrm{tp},\mathrm{Gag}}x_{16T}$	96	$x_{27T}^{--}$	$d_{\mathrm{free}}x_{27T}$
17	$x_{4}^{--},x_{5}^{++}$	$k_{{\mathrm{DNA}}_{t}}x_{4}$	57	$x_{16T}^{--}$	$d_{\mathrm{Gag}}x_{16}$	97	$x_{27\mathrm{HT}}^{--}$	$d_{\mathrm{free}}x_{27\mathrm{HT}}$
18	$x_{4}^{--}$	$d_{{\mathrm{DNA}}_{\mathrm{cor}}}x_{4}$	58	$x_{17}^{++}$	$k_{\mathrm{trans}}f_{\mathrm{gp}160}x_{11}$	98	$x_{28}^{++}$	$k_{\mathrm{RIG}1}\left(x_{3}+x_{3T}\right)$
19	$x_{4T}^{--},x_{5T}^{++}$	$k_{{\mathrm{DNA}}_{t}}x_{4T}$	59	$x_{17}^{--},x_{22}^{++}$	$k_{\mathrm{tp},\mathrm{gp}160}x_{17}$	99	$x_{28}^{--}$	$d_{\mathrm{RIG}}x_{28}$
20	$x_{4T}^{--}$	$d_{{\mathrm{DNA}}_{\mathrm{cor}}}x_{4T}$	60	$x_{17}^{--}$	$d_{\mathrm{gp}160}x_{10}$	100	$x_{29}^{++}$	$k_{\mathrm{IRF}3}x_{28}$
21	$x_{5}^{--},x_{6}^{++}$	$k_{\mathrm{int}}x_{5}$	61	$x_{18}^{++}$	$k_{\mathrm{trans}}f_{\mathrm{ss},\mathrm{Vpu}}x_{11}$	101	$x_{29}^{--}$	$d_{\mathrm{IRF}3}x_{29}$
22	$x_{5}^{--}$	$d_{{\mathrm{DNA}}_{\mathrm{nuc}}}x_{5}$	62	$x_{18}^{--},x_{24}^{++}$	$k_{\mathrm{tp}}x_{18}$	102	$x_{30}^{++}$	$k_{\mathrm{NF}-\mathrm{kb}}x_{28}$
23	$x_{5T}^{--},x_{6T}^{++}$	$k_{\mathrm{int}}x_{5T}$	63	$x_{18}^{--}$	$d_{\mathrm{Vpu}}x_{18}$	103	$x_{30}^{--}$	$d_{\mathrm{NF}-\mathrm{kb}}x_{30}$
24	$x_{5T}^{--}$	$d_{{\mathrm{DNA}}_{\mathrm{nuc}}}x_{5T}$	64	$x_{19}^{++}$	$k_{\mathrm{trans}}f_{\mathrm{ss},\mathrm{Vif}}x_{11}$	104	$x_{31}^{++}$	$k_{\mathrm{IFNi}}x_{29}$
25	$x_{6}^{--}$	$d_{{\mathrm{DNA}}_{\mathrm{cor}}}x_{6}$	65	$x_{19}^{--}$	$d_{\mathrm{Vif}}x_{19}$	105	$x_{31}^{++}$	$k_{\mathrm{IFNi}}x_{30}$
26	$x_{6T}^{--}$	$d_{{\mathrm{DNA}}_{\mathrm{cor}}}x_{6T}$	66	$x_{20}^{--}$	$k_{\mathrm{comb}}N_{\mathrm{GagPol}}\left(\Theta_{c}+\Theta_{c}^{T}+\Theta_{c}^{\mathrm{HT}}\right)$	106	$x_{31}^{--},x_{32}^{++}$	$k_{e}x_{31}$
27	$x_{7}^{++}$	$f_{\mathrm{TR}}x_{6}$	67	$x_{20}^{--}$	$d_{\mathrm{mem},\mathrm{GagPol}}x_{20}$	107	$x_{31}^{--}$	$d_{\mathrm{IFNi}}x_{31}$
28	$x_{7}^{--},x_{8}^{++}$	$k_{\mathrm{eRNA}}f_{\mathrm{Rev}}x_{7}$	68	$x_{21}^{--}$	$k_{\mathrm{comb}}N_{\mathrm{Gag}}\left(\Theta_{c}+\Theta_{c}^{T}+\Theta_{c}^{\mathrm{HT}}\right)$	108	$x_{32}^{--}$	$d_{\mathrm{IFNe}}x_{32}$
29	$x_{7}^{--},x_{10}^{++}$	$k_{{\mathrm{ssRNA}}_{g}}g_{\mathrm{Rev}}x_{7}$	69	$x_{21}^{--}$	$d_{\mathrm{mem},\mathrm{Gag}}x_{21}$	109	$x_{33}^{++}$	$k_{\mathrm{STAT}}x_{32}$
30	$x_{7}^{--}$	$d_{{\mathrm{DNA}}_{\mathrm{cor}}}x_{7}$	70	$x_{22}^{--}$	$k_{\mathrm{comb}}N_{\mathrm{gp}160}\left(\Theta_{c}+\Theta_{c}^{T}+\Theta_{c}^{\mathrm{HT}}\right)$	110	$x_{33}^{--}$	$d_{\mathrm{STAT}}x_{33}$
31	$x_{7T}^{++}$	$f_{\mathrm{TR}}x_{6T}$	71	$x_{22}^{--}$	$d_{\mathrm{mem},\mathrm{gp}160}x_{22}$	111	$x_{34}^{++}$	$k_{\mathrm{ISG}}x_{33}$
32	$x_{7T}^{--},x_{10T}^{++}$	$k_{{\mathrm{eRNA}}_{g}}g_{\mathrm{Rev}}x_{7T}$	72	$x_{23}^{--}$	$k_{\mathrm{comb}}\left(N_{\mathrm{RNA}}\Theta_{c}+\Theta_{c}^{\mathrm{HT}}\right)$	112	$x_{34}^{--}$	$f_{\mathrm{Vif}}x_{20}x_{34}$
33	$x_{7T}^{--}$	$d_{{\mathrm{DNA}}_{\mathrm{cor}}}x_{7T}$	73	$x_{23}^{--}$	$d_{\mathrm{RNA}}x_{23}$	113	$x_{34}^{--}$	$d_{\mathrm{APO}}x_{34}$
34	$x_{8}^{--},x_{9}^{++}$	$k_{{\mathrm{dsRNA}}_{\mathrm{ss}}}g_{\mathrm{Rev}}x_{8}$	74	$x_{23T}^{--}$	$k_{\mathrm{comb}}\left(N_{\mathrm{RNA}}\Theta_{c}^{T}+\Theta_{c}^{\mathrm{HT}}\right)$	114	$x_{35}^{++}$	$k_{\mathrm{ISG}}x_{33}$
35	$x_{8}^{--},x_{11}^{++}$	$k_{{\mathrm{eRNA}}_{\mathrm{ss}}}g_{\mathrm{Rev}}x_{8}$	75	$x_{23T}^{--}$	$d_{\mathrm{RNA}}x_{23T}$	115	$x_{35}^{--}$	$d_{\mathrm{SAMHD}1}x_{35}$
36	$x_{8}^{--}$	$d_{{\mathrm{RNA}}_{\mathrm{ss}}}x_{8}$	76	$x_{24}^{--}$	$d_{\mathrm{Vpu}}x_{24}$	116	$x_{36}^{++}$	$k_{\mathrm{ISG}}x_{33}$
37	$x_{9}^{--},x_{12}^{++}$	$k_{{\mathrm{eRNA}}_{\mathrm{ds}}}g_{\mathrm{Rev}}x_{8}$	77	$x_{25}^{++}$	$k_{\mathrm{comb}}\Theta_{c}$	117	$x_{36}^{--}$	$f_{\mathrm{Vpu}}x_{24}x_{36}$
38	$x_{9}^{--}$	$d_{{\mathrm{RNA}}_{\mathrm{ds}}}x_{9}$	78	$x_{25}^{--},x_{26}^{++}$	$k_{\mathrm{bud}}x_{25}$	118	$x_{36}^{--}$	$d_{\mathrm{Teth}}x_{36}$
39	$x_{10}^{--},x_{23}^{++}$	$k_{\mathrm{tp},\mathrm{RNA}}x_{10}$	79	$x_{25}^{--}$	$d_{\mathrm{comb}}x_{25}$	–		
40	$x_{10}^{--}$	$d_{{\mathrm{RNA}}_{g}}x_{10}$	80	$x_{25T}^{++}$	$k_{\mathrm{comb}}\Theta_{c}^{T}$	–		

**Table 2 viruses-17-01378-t002:** Values of parameter *S* defined in Equation (Equation 66) versus MOI for different numbers of extracellular molecules of IFN.

MOI:	1	2	3	4	5	6	7	8	9	10
IFN = 0	0.405	0.674	0.865	0.995	1.091	1.160	1.224	1.281	1.331	1.374
IFN = 5	0.287	0.478	0.629	0.750	0.850	0.916	0.990	1.047	1.104	1.154
IFN = 10	0.196	0.348	0.471	0.582	0.663	0.735	0.797	0.857	0.905	0.961

## Data Availability

The original contributions presented in this study are included in the article. Further inquiries can be directed to the corresponding authors.

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
