# Peer review of "Quantifying the Inhibitory Efficacy of HIV-1 Therapeutic Interfering Particles at a Single CD4 T-Cell Resolution"

_viruses, 2025, doi:10.3390/v17101378_

Round 1
Reviewer 1 Report
Comments and Suggestions for Authors
Defective interfering particles have long been known to have an inhibitory effect on the growth of wild type viruses from many different genera. For HIV, there is growing interest in the field, as the paper in Science by Leor Weinberger and colleagues in 2024 provided exciting evidence that rationally designed HIV-1 TIPs could drive significant (~3 log) reductions in viral loads in animal models with no signs of viral escape. This manuscript attempts to provide quantitative clarity on the mechanism of action of HIV-1 TIPs. Through mathematical modeling the authors conclude that TIPs alone are relatively inefficient inhibitors of wild-type virus particle formation and infection but that in the presence of modest levels of type I IFN they synergize to dramatically suppress infectious virus replication and spread.
All in all, this is a valuable study that suggests that TIP “competition” with wild-type virus is likely less effective at reducing wild-type virus replication in vivo than originally speculated, but that other host-mediated components are necessary to boost TIP efficacy into a range where a clear phenotype can be observed. This opens the door to some potential modifications to TIPs that may enhance induction of type I IFN or other immune modulators and give these intriguing agents enhanced therapeutic power for HIV-1 and beyond.
Comments:
- It is unclear if the TIPs the authors use in their modeling induce a similar pattern of type I IFN as wild-type virus. The authors should put their data in a more physiological context of what is known about type I IFN induction during HIV-1 infection and the mechanisms the virus uses to short-circuit type I IFN production/activity. This would be helpful to add to the discussion and move the conclusions beyond the theoretical.
- Generally, DI particles in viral systems are very good inhibitors during initial infections, but lose effectiveness once titers decrease, allowing wild type virus to exponentially expand (along with DI particles). Further inhibition by DI particles can occur, but generally they are lost over time. Can the authors comment on this behavior as it pertains to their model?
- One note: there are several grammatical errors in the paper that should be corrected prior to publication.
Author Response
Response to comments of Reviewer 1
Defective interfering particles have long been known to have an inhibitory effect on the growth of wild type viruses from many different genera. For HIV, there is growing interest in the field, as the paper in Science by Leor Weinberger and colleagues in 2024 provided exciting evidence that rationally designed HIV-1 TIPs could drive significant (~3 log) reductions in viral loads in animal models with no signs of viral escape. This manuscript attempts to provide quantitative clarity on the mechanism of action of HIV-1 TIPs. Through mathematical modeling the authors conclude that TIPs alone are relatively inefficient inhibitors of wild-type virus particle formation and infection but that in the presence of modest levels of type I IFN they synergize to dramatically suppress infectious virus replication and spread.
All in all, this is a valuable study that suggests that TIP “competition” with wild-type virus is likely less effective at reducing wild-type virus replication in vivo than originally speculated, but that other host-mediated components are necessary to boost TIP efficacy into a range where a clear phenotype can be observed. This opens the door to some potential modifications to TIPs that may enhance induction of type I IFN or other immune modulators and give these intriguing agents enhanced therapeutic power for HIV-1 and beyond.
We thank the Reviewer for insightful comments and the thorough work on our manuscript.
There are a few points that I want to mention and I think the authors should address these very carefully.
- It is unclear if the TIPs the authors use in their modeling induce a similar pattern of type I IFN as wild-type virus. The authors should put their data in a more physiological context of what is known about type I IFN induction during HIV-1 infection and the mechanisms the virus uses to short-circuit type I IFN production/activity. This would be helpful to add to the discussion and move the conclusions beyond the theoretical.
In response to the suggestion, we have added the following text to Discussion section.
Type I interferon proteins (IFNs) are an important component of cellular antiviral defense mechanisms. They are activated after virus infections through specialized pattern recognition receptors that recognize viral RNAs or DNAs within different cellular locations. The produced IFNs then activate expression of IFN-stimulated genes in infected cells and in surrounding cells and trigger an antiviral state that reduces virus production. Infections with HIV-1 are no exceptions and IFNs are produced throughout the entire infection course [1]. The TIPs described by Pitchai et al. [2] and studied here contain structural HIV-1 elements like the TAR stem loop structure and the U-rich region of the LTR in their genomes that are key inducers of the IFN-I response [3]. Furthermore, in TIP-treated animals ([2] Supplementary materials, fig. S14), qRT-PCR analysis of RNAs from retropharyngeal lymph node tissue indicated no significant differences in the expression of selected inflammatory cytokines, including IFN-alpha and IFN-beta, between TIP-treated and control animals. Hence, it seems justified to assume in our developed model that TIPs induce a similar pattern of type I IFN responses as wild-type HIV-1.
[1]. Mackelprang RD, Filali-Mouhim A, Richardson B, et al. Upregulation of IFN-stimulated genes persists beyond the transitory broad immunologic changes of acute HIV-1 infection. iScience. 2023;26(4):106454. Published 2023 Mar 21. doi:10.1016/j.isci.2023.106454
[2]. Pitchai FNN, Tanner EJ, Khetan N. Vasen G, Levrel C, Kumar AJ, Pandey S, Ordonez T, Barnette P, Spencer D. et al. Engineered deletions of HIV replicate conditionally to reduce disease in nonhuman primates. Science 2024, 385, eadn5866. 505 https://doi.org/10.1126/science.adn5866.
[3]. Berg RK, Melchjorsen J, Rintahaka J, et al. Genomic HIV RNA induces innate immune responses through RIG-I-dependent sensing of secondary-structured RNA. PLoS One. 2012;7(1):e29291. doi:10.1371/journal.pone.0029291
- Generally, DI particles in viral systems are very good inhibitors during initial infections, but lose effectiveness once titers decrease, allowing wild type virus to exponentially expand (along with DI particles). Further inhibition by DI particles can occur, but generally they are lost over time. Can the authors comment on this behavior as it pertains to their model?.
In response to the comment, we have added the following text to Discussion section.
“Our model is formulated to describe a single round of HIV-1 replication in the presence of TIP-2. The effect of co-infection with TIP-2 on production of infectious HIV-1 in our study is characterized by two integrative parameters, i.e. the TIP-2 inhibitory factor and Life Cycle Efficiency presented in Figures 17 and 18. The model predicts that both characteristics depend on the MOI and reduce their values with a decrease in the number of HIV-1 infecting the cell. Overall, it implies that a reduction in viral load in patients under HAART could be associated with a reduced effectiveness of TIP-2 over time of infection.”.
This behavior is consistent with the above mentioned reduction of DI effectiveness once viral titers decrease.
- One note: there are several grammatical errors in the paper that should be corrected prior to publication.
All found grammatical errors have been corrected.

Reviewer 2 Report
Comments and Suggestions for Authors
The current study is based on the controversial conclusion from two published literatures. On top of it, the authors have generated a mathematical model that describes the interference of TIP with HIV replication within infected CD4+ T cells. Overall, the model can be used for designing combination therapy for HIV cure. I believe the study is helpful in general, however, I feel it really takes time and patience to read and catch the point. Because there are equations almost every single where, which becomes my biggest concern. It would be great if the authors can change the way of writing, trying to make the manuscript straightforward and understandable. Below are my comments and questions to help improve the quality of publication.
- There are large repetitive content shows between Abstract and Conclusion. They are exactly the same sentences showing at two places. Please re-write the Conclusion or delete it since this section is not mandatory according to the journal guideline.
- For reference format, please remove the extra online link and only keep the doi. link. Taking reference 2 as an example, please remove the link of [https://onlinelibrary.wiley.com/doi/pdf/10.1002/hsr2.70089] since it directs the reader to nowhere and keep the doi. link of https://doi.org/https://doi.org/10.1002/hsr2.70089. Please check them all carefully throughout the Reference.
- The first paragraph of Introduction describes the efficient control of HIV. There is a recently published review of “Development of anti-HIV therapeutics: from conventional drug discovery to cutting-edge technology”, it summarizes the conventional ART drugs as well as novel strategies of HIV treatment, such as CRIPSR and CRISPR base editing. I would suggest the authors to include this literature here to make the content more completed.
- Line 51, the sentence is not completed in expression. “The TIP genomic structure is determined by the deletion of the trans elements as in [8].” Please update it.
- Line 66-68, please rephrase the sentence “We consider the replication…….expression [8]”. It has grammar mistake.
- I suggest combining Figure 1 and 2 to show HIV and TIP replication in one figure. It will be much easier for the readers to understand both schemes.
- There are massive equations in the manuscript, which I feel so hard to read and understand. Is there anyway to make it concise and simple? For example, table 1 is filled of tons of equations, but I am not sure what information I can get from it.
There are lots of long sentences, which makes it hard to read and understand. Some grammar mistakes exist in the manuscript. I would recommend the service of language editing for publication.
Author Response
Response to comments of Reviewer 2
The current study is based on the controversial conclusion from two published literatures. On top of it, the authors have generated a mathematical model that describes the interference of TIP with HIV replication within infected CD4+ T cells. Overall, the model can be used for designing combination therapy for HIV cure. I believe the study is helpful in general, however, I feel it really takes time and patience to read and catch the point. Because there are equations almost every single where, which becomes my biggest concern. It would be great if the authors can change the way of writing, trying to make the manuscript straightforward and understandable.
We thank the Reviewer for insightful comments and the thorough work on our manuscript.
- There are large repetitive content shows between Abstract and Conclusion. They are exactly the same sentences showing at two places. Please re-write the Conclusion or delete it since this section is not mandatory according to the journal guideline.
In response to the suggestion, we have deleted the Conclusion section.
- For reference format, please remove the extra online link and only keep the doi. link. Taking reference 2 as an example, please remove the link of [https://onlinelibrary.wiley.com/doi/pdf/10.1002/hsr2.70089] since it directs the reader to nowhere and keep the doi. link of https://doi.org/https://doi.org/10.1002/hsr2.70089. Please check them all carefully throughout the Reference.
Corrected.
- The first paragraph of Introduction describes the efficient control of HIV. There is a recently published review of “Development of anti-HIV therapeutics: from conventional drug discovery to cutting-edge technology”, it summarizes the conventional ART drugs as well as novel strategies of HIV treatment, such as CRIPSR and CRISPR base editing. I would suggest the authors to include this literature here to make the content more completed.
This reference has been inserted. The new text reads as follows
Other ways to control viral replication are under investigation. These include:
- HIV vaccination \cite{Scott2024},
- preexposure prophylaxis \cite{MAYER2022257},
- combination therapies \cite{Landovitz2023},
- novel strategies such as CRISPR and CRISPR-base editing \cite{Sun2024},
- defective interfering particles (DIPs) which can be genetically engineered to become therapeutic interfering particle (TIPs) \cite{Tanner2016}.
- Line 51, the sentence is not completed in expression. “The TIP genomic structure is determined by the deletion of the trans elements as in [8].” Please update it.
The sentence has been updated and reads as follows.
“The TIP genomic structure is determined by a deletion of the trans-elements, i.e. a 2.5-kb deletion in the HIV pol-vpr region, and the reintroduction of the central polypurin tract (cPPT) as described in [8].”
- Line 66-68, please rephrase the sentence “We consider the replication…….expression [8]”. It has grammar mistake.
The sentence has been corrected and reads as follows.
We examine the replication of engineered TIP-2 particles that differ from HIV-1 by (1) a 2.5-kb deletion in the HIV pol-vpr region, (2) ablation of the coding sequences of tat/rev/vpu/env, and (3) additional deletions in gag and env as detailed in \cite{Weinberger2024}.
- I suggest combining Figure 1 and 2 to show HIV and TIP replication in one figure. It will be much easier for the readers to understand both schemes.
We intendedly put the replication schemes of HIV, TIP and heterozygote HIV-TIP in three different pictures to make them more readable. We were afraid that after combining all components in a single scheme would be too difficult to understand. However, we now locate figures 1 and 2 on the same page to simplify the perception of the schemes.
- There are massive equations in the manuscript, which I feel so hard to read and understand. Is there anyway to make it concise and simple? For example, table 1 is filled of tons of equations, but I am not sure what information I can get from it.
Our work is dealing with mathematical and numerical modelling of viral dynamics. The number of equations is defined by the number of components to be accounted. Equations (1)-(57) describe the deterministic model. Equations in Table 1 describe the stochastic model in terms of Markov chain.
Comments on the Quality of English Language
There are lots of long sentences, which makes it hard to read and understand. Some grammar mistakes exist in the manuscript. I would recommend the service of language editing for publication.
All found grammatical errors have been corrected. Long sentences have been edited.

Reviewer 3 Report
Comments and Suggestions for Authors
Authors
- A thorough revision of the English is strongly recommended. There are numerous grammatical and spelling errors that need to be corrected. I have noted some of them in my minor comments, but many more remain and should be addressed systematically.
Abstract
- The abstract highlights success in primates, but the Introduction presents a more compelling contrast: success in primates versus lack of effect in humans. Including this discrepancy in the abstract would increase its impact.
- The term TIP-2 appears without context. Readers may wonder what TIP-2 is or why the numbering matters. A short clarification would help, e.g. “a specific engineered particle, TIP-2…”.
- IFN-I is used at the end of the abstract but never defined. Please define all abbreviations at first mention.
- The phrase “enable to reduce” is ungrammatical. Prefer “…that enable the reduction of…” or simply “…that reduce…”.
- Clarify what “lower viral production” means in the 1:10 ratio scenario, by how much?
Introduction
- The introduction describes the discrepancy between primate studies (engineered TIPs) and human studies (natural DIPs). This distinction underlies the hypothesis for differing outcomes. Making this hypothesis more explicit would strengthen the rationale.
- The model focuses on TIP-2 from 10.1126/science.adn5866, but this term only appears in Methods. It would be more coherent to introduce it earlier.
- TIPs are said to interfere with HIV replication, but the mechanism is not clearly specified. Literature describes several possible mechanisms (competition for cellular/viral factors such as Gag, competitive RNA encapsidation, or non-viable heterodimers). The model necessarily assumes one or more of these. Which is considered most relevant?
- Prior in silico work [7] suggested that heterodimers were “evolutionarily unstable.” The current model includes heterodimer production. How do the authors reconcile this? Since heterodimerization allows recombination, there is a nontrivial safety risk: HIV could recombine with TIP genomes, potentially restoring lost functions or generating new variants. This risk should be acknowledged in the introduction.
- TIPs, as viral genomes, are recognized by innate immune sensors (RIG-I), inducing IFN responses. Thus, antiviral effects may not only arise from direct competition but also from host innate immunity. This possibility should be addressed.
- The model analyzes single-cell dynamics, yet therapy efficacy at patient level depends on population dynamics—spread of TIPs vs HIV across cells. This limitation should be noted.
M&M
- Section 2.1 is too schematic and does not bridge biology to mathematics. For instance, TIP-2 has a 2.5 kb deletion eliminating tat/rev/vpu/env. How is this represented in the equations? A comparison table would be very helpful, e.g. viral life cycle stage, HIV-1 model component, TIP-2 model component, and rationale (based on deletion).
- Section 2.2: Parameter derivation lacks transparency. HIV-1 parameters are cited appropriately, but how were TIP-2 parameters obtained? A separate table should list TIP-2–specific or modified parameters. Some equations (e.g. assembly, Fc, θ regulation) are mathematically complex but not explained qualitatively. A short explanatory paragraph would improve clarity and reproducibility.
- Section 2.4: The hybrid model implementation is poorly described. Key details (e.g. the X threshold) are missing. A dedicated paragraph explaining the algorithm, threshold values, rationale, and significance is needed.
Results
- Figures 11-18 compare multiple conditions (MOI, TIPs, IFN). Statements such as “reduces production” or “increases production” need quantification and confidence.
- Section 3.4 describes multimodal histograms (Fig. 9) but does not interpret them. Likely, each peak corresponds to discrete numbers of integrated viral/TIP genomes. This biological interpretation is important.
- Figure 15, showing relative amplification (TIP vs HIV), is a key finding. The concept is introduced but not fully explored. It defines when TIPs “outcompete” HIV, this should be emphasized as the therapeutic goal.
- New metrics (LCE, IF) are introduced but not combined into a practical “therapeutic efficacy map.” A heatmap or phase diagram would summarize key results: X-axis = initial TIPs, Y-axis = initial MOI, color = probability of functional extinction (<1 HIV virion). Separate maps could be shown for each IFN condition. Time-course plots of IF under different scenarios would also clarify therapeutic windows.
Discussion/Conclusions
- The synergy between IFN and TIPs could explain success in primates. Could species-specific factors (SHIV vs HIV, acute vs chronic infection, baseline immunity) account for this? This deserves discussion.
- At low MOI, IFN may reduce total viral load but also decrease TIP’s relative advantage. Does this mean TIP therapy could be less effective under ART-suppressed conditions when combined with IFN stimulation?
- Lines 554-557 mention recombination risk. Please expand on the potential consequences based on known HIV molecular biology.
- Add a paragraph of limitations: single-cell model, parameter assumptions, homogeneous cell population, lack of adaptive immunity modeling, etc.
- The conclusion should be more concise and forward-looking. Instead of repeating details, highlight the main finding, therapeutic implication (TIPs as immune sensitizers or combination partners, not monotherapy), contribution of the model (rational design tool), and an outlook toward cure.
Minor
- Replace “non-humate primates” with “non-human primates”
- Abstract: “The model predict” by “The model predicts”
- Abstract: “reduction of HIV-infection” by “reduction of HIV infection”
- Introduction: “defective interfering particle (TIPs)” by “defective interfering particles (TIPs)”
- L29: “HIV is dipliod” by “HIV is diploid”
- L43: Clarify “this interference” (currently ambiguous)
- L51: “…a parallel set of equations the TIP-2 replication…” by “…a parallel set of equations for the TIP-2 replication…”
- L150: “In HIV-1 life cycle, he encoded messenger RNAs…” by “In the HIV-1 life cycle, the encoded messenger RNAs…”
- L449: “…cell incrasese with…” by “…cell increases with…”
- L450: “However. the extent…” by “However, the extent…”
- While the abbreviation list is useful, ensure all acronyms (MOI, ISGs, ODE, MC, IFN-I in the abstract, etc.) are defined at first use.
Author Response
Response to comments of Reviewer 3
We thank the Reviewer for insightful comments and the thorough work on our manuscript.
Abstract
- The abstract highlights success in primates, but the Introduction presents a more compelling contrast: success in primates versus lack of effect in humans. Including this discrepancy in the abstract would increase its impact.
To address this question, we have added the following text to Abstract:
“However, human studies showed that no beneficial effect of defective interfering particles (DIPs) in people living with HIV-1 \cite{Hariharan2025}. The discrepancy between the two studies highlights the importance to further investigate HIV-TIP interactions.”
- The term TIP-2 appears without context. Readers may wonder what TIP-2 is or why the numbering matters. A short clarification would help, e.g. “a specific engineered particle, TIP-2…”?
To address this question, we have modified the respective sentence in Abstract to read:
“In our study we developed a high-resolution mathematical model to study various aspects of the interference of a specific engineered particle, TIP-2, characterized by a 2.5-kb deletion in the HIV \textit{ pol-vpr} region, with HIV-1 replication within infected CD4+ T cells”.
- IFN-I is used at the end of the abstract but never defined. Please define all abbreviations at first mention.
Corrected.
- The phrase “enable to reduce” is ungrammatical. Prefer “…that enable the reduction of…” or simply “…that reduce…”.
Corrected. The new text reads:
“We define the conditions in terms of the number of homozygous HIV-1 virions and TIP-2 particles that enable the reduction of the wild-type virus replication rate to about one.”
- Clarify what “lower viral production” means in the 1:10 ratio scenario, by how much?.
To address this question, we have added the number as predicted by the ODE model:
“The model predicts that at a ratio of 1 HIV-1 to 10 TIP-2 particles, the infected cell still produces some viruses although in a minor quantity, i.e. about two viruses per cycle.”
Introduction
- The introduction describes the discrepancy between primate studies (engineered TIPs) and human studies (natural DIPs). This distinction underlies the hypothesis for differing outcomes. Making this hypothesis more explicit would strengthen the rationale.
In response to the suggestion, we have added the following sentence:
“The distinction between the structure of engineered TIPs and the naturally emerging DIPs in humans is an important factor contributing to the differing outcomes of the competition between HIV-1 and TIPs or DIPs in primate studies and in humans, respectively.”
- The model focuses on TIP-2 from 10.1126/science.adn5866, but this term only appears in Methods. It would be more coherent to introduce it earlier.
In response to this suggestion, we have added the following text to Introduction:
“The TIP-2 genomic structure is determined by a deletion of the trans-elements, i.e. a 2.5-kb deletion in the HIV pol-vpr region, and the reintroduction of the central polypurin tract (cPPT) as described in \cite{Weinberger2024}.”
- TIPs are said to interfere with HIV replication, but the mechanism is not clearly specified. Literature describes several possible mechanisms (or non-viable heterodimers). The model necessarily assumes one or more of these. Which is considered most relevant?
In response to this suggestion, we have added the following text to subsection 2.1:
“As follows from the replication cycle of HIV-1 and TIP-2 and their genomic structures, see Figures 1 and 2, the model assumes that the viruses and TIPs compete for the viral proteins Gag-Pol, Gag and gp-160.”
- Prior in silico work [7] suggested that heterodimers were “evolutionarily unstable.” The current model includes heterodimer production. How do the authors reconcile this? Since heterodimerization allows recombination, there is a nontrivial safety risk: HIV could recombine with TIP genomes, potentially restoring lost functions or generating new variants. This risk should be acknowledged in the introduction.
In response to this remark, we have added the following text to Introduction:
“Since heterodimerization enables recombination, there is a nontrivial safety risk as HIV could recombine with TIP genomes, potentially restoring lost functions or generating new variants. Hence, the kinetics of heterozygous HIV-TIP generation requires further examination.”
- TIPs, as viral genomes, are recognized by innate immune sensors (RIG-I), inducing IFN responses. Thus, antiviral effects may not only arise from direct competition but also from host innate immunity. This possibility should be addressed.
In response to this question, we have added the following commentary to Discussion
“Type I interferon proteins (IFNs) are an important component of cellular antiviral defense mechanisms. They are activated after virus infections through specialized pattern recognition receptors that recognize viral RNAs or DNAs within different cellular locations. The produced IFNs then activate expression of IFN-stimulated genes in infected cells and in surrounding cells and trigger an antiviral state that reduces virus production. Infections with HIV-1 are no exceptions and IFNs are produced throughout the entire infection course [1]. The TIPs described by Pitchai et al. [2] and studied here contain structural HIV-1 elements like the TAR stem loop structure and the U-rich region of the LTR in their genomes that are key inducers of the IFN-I response [3]. Furthermore, in TIP-treated animals ([2] Supplementary materials, fig. S14), qRT-PCR analysis of RNAs from retropharyngeal lymph node tissue indicated no significant differences in the expression of selected inflammatory cytokines, including IFN-alpha and IFN-beta, between TIP-treated and control animals. Hence, it seems justified to assume in our developed model that TIPs induce a similar pattern of type I IFN responses as wild-type HIV-1.”
[1]. Mackelprang RD, Filali-Mouhim A, Richardson B, et al. Upregulation of IFN-stimulated genes persists beyond the transitory broad immunologic changes of acute HIV-1 infection. iScience. 2023;26(4):106454. Published 2023 Mar 21. doi:10.1016/j.isci.2023.106454
[2]. Pitchai FNN, Tanner EJ, Khetan N. Vasen G, Levrel C, Kumar AJ, Pandey S, Ordonez T, Barnette P, Spencer D. et al. Engineered deletions of HIV replicate conditionally to reduce disease in nonhuman primates. Science 2024, 385, eadn5866. 505 https://doi.org/10.1126/science.adn5866.
[3]. Berg RK, Melchjorsen J, Rintahaka J, et al. Genomic HIV RNA induces innate immune responses through RIG-I-dependent sensing of secondary-structured RNA. PLoS One. 2012;7(1):e29291. doi:10.1371/journal.pone.0029291
- The model analyzes single-cell dynamics, yet therapy efficacy at patient level depends on population dynamics—spread of TIPs vs HIV across cells. This limitation should be noted.
In response to this suggestion, we have added the following text to the end of Discussion.
“The presented model analyzes the HIV and TIP replication cycles at a single-cell level. However, the efficacy of the TIP-based therapy at patient level depends on population dynamics, i.e. the spread of TIPs vs HIV across target cells. The extension of the model from a single cell to a tissue level will be a necessary step for translation of the model findings to clinically relevant predictions.”
M&M
- Section 2.1 is too schematic and does not bridge biology to mathematics. For instance, TIP-2 has a 2.5 kb deletion eliminating tat/rev/vpu/env. How is this represented in the equations? A comparison table would be very helpful, e.g. viral life cycle stage, HIV-1 model component, TIP-2 model component, and rationale (based on deletion).
In response to the suggestion of the reviewer, it is explained now at the beginning of the Transcription-splicing paragraph and in the sentence following equations (21) and (22).
“The difference in genomic organization of HIV and TIP-2 reveals at the RNA transcription stage. The Tat-dependent regulation of the transcription rate takes into account that TIP-2 has a shorter nucleotide length.”
- Section 2.2: Parameter derivation lacks transparency. HIV-1 parameters are cited appropriately, but how were TIP-2 parameters obtained? A separate table should list TIP-2–specific or modified parameters. Some equations (e.g. assembly, Fc, θ regulation) are mathematically complex but not explained qualitatively. A short explanatory paragraph would improve clarity and reproducibility.
In response to the suggestion of the reviewer, we have now added the following paragraph at the end of subsection 2.1.
“The parameters of the biochemical reactions underlying the HIV-1 and TIP-2 replication stages are assumed to be the same except for the RNA transcription and splicing rates which are corrected for the shorter length of TIP-2. To describe the rate limiting reactions representing the Tat-Rev-dependent regulation mRNA abundance, a Michaelis-Menten type parameterization is used. For the description of the assembly of pre-virions, pre-TIP-2 and pre-HIV-TIP-2 particles, we assume that the dependence of the assembly rate on Gag-Pol, Gag and gp-160 saturates at high levels of these proteins. The dependence of the assembly rate on the number of RNA molecules follows a second order kinetics due to the diploid nature of the viral genome but saturates at high densities of the genomic RNAs.”
- Section 2.4: The hybrid model implementation is poorly described. Key details (e.g. the X threshold) are missing. A dedicated paragraph explaining the algorithm, threshold values, rationale, and significance is needed.
It is described thoroughly in our previous works [11,13]. Nevertheless, we explained it in more details in the revised version close to the end of subsection 2.4.
“In this algorithm, the dynamics for any component, $x_n$, automatically switches from stochastic to deterministic once its abundance exceeds a predefined threshold $\bar{X}$. If the abundance later falls below $\bar{X}$, it switches back to stochastic dynamics with rounding the population to an integer value. Thus, the stochastic and deterministic processes for different components are performed in parallel. We set a threshold of $\bar{X} = 10^4$ for all components; only a few protein species reached this level.
Comparisons of statistic characteristics: mean, median, etc., computed by the fully stochastic and hybrid models showed negligible discrepancy with this threshold, whereas the use of hybrid scheme reduces computation time by a factor of four.”
Results
- Figures 11-18 compare multiple conditions (MOI, TIPs, IFN). Statements such as “reduces production” or “increases production” need quantification and confidence.
We think that the quantification of reduction/growth of production is seen from the plots presented in Figures 11-18. It would take a lot of space to describe all the cases in numbers.
As for the confidence of the computed results we have to make several remarks:
(i) Adding the confidence intervals to the plots would make them messy, because of a significant number of plotted points and their density.
(ii) Because of the big size of the sample (104 realizations) used for every point, the confidence intervals would be too narrow: close or smaller than the size of markers pictured in the plot.
(iii) The smoothness of curves indicates the robustness of the computed results: the confidence intervals would be wider for the smaller sample size, the plotted points would be more scattered, and the plotted curves would be less smooth.
- Section 3.4 describes multimodal histograms (Fig. 9) but does not interpret them. Likely, each peak corresponds to discrete numbers of integrated viral/TIP genomes. This biological interpretation is important.
In response to the suggestion of the Reviewer, we have added the followings sentence.
“Each peak of the histograms corresponds to discrete numbers of integrated viral/TIP genomes observed in real HIV infection [1].”
[1]. Jung A, Maier R, Vartanian JP, et al. Recombination: Multiply infected spleen cells in HIV patients. Nature. 2002;418(6894):144. doi:10.1038/418144a
- Figure 15, showing relative amplification (TIP vs HIV), is a key finding. The concept is introduced but not fully explored. It defines when TIPs “outcompete” HIV, this should be emphasized as the therapeutic goal.
In response to the suggestion we have now added the following comment at the end of section 3.6.
“Overall, the model predicts that the competition between TIP-2 and HIV-1 depends on the multiplicity of infection and the activity of the IFN-I system. As the major therapeutic goal of TIP-2 application is to outcompete HIV-production, care should be given to quantitative levels of MOI and IFN-I in designing the protocols of TIP-2-based treatments.”
- New metrics (LCE, IF) are introduced but not combined into a practical “therapeutic efficacy map.” A heatmap or phase diagram would summarize key results: X-axis = initial TIPs, Y-axis = initial MOI, color = probability of functional extinction (<1 HIV virion). Separate maps could be shown for each IFN condition. Time-course plots of IF under different scenarios would also clarify therapeutic windows.
As for probability of functional extinction, it is plotted for various combinations of MOI, TIP-2 and IFN-I in Figures 19,20. We did not use the colored maps as the current plot types give in our view a clearer quantitative information about the value of the probability. The IF factor characterizes the overall efficacy for one replication cycle rather than its time-course.
Discussion/Conclusions
- The synergy between IFN and TIPs could explain success in primates. Could species-specific factors (SHIV vs HIV, acute vs chronic infection, baseline immunity) account for this? This deserves discussion.
To address the above question theoretically, one would need to extend the single cell model to a systemic level description of HIV and SHIV infections which is beyond our study. We have now added the respective statement to Discussion.
“The model-predicted synergy between IFN-I and TIPs could be one factor explaining the success of TIP-2-based therapy in primates. However, other species-specific factors such as SHIV vs HIV, acute vs chronic infection and baseline immunity need to be also taken into account. To address the above question theoretically, one would need a further extension of the presented single cell model to a systemic level description of HIV and SHIV infections.”
- At low MOI, IFN may reduce total viral load but also decrease TIP’s relative advantage. Does this mean TIP therapy could be less effective under ART-suppressed conditions when combined with IFN stimulation?
In response to the above suggestion, we have added the following text to Discussion.
“However, the model predicts that the outcome of competition between TIP-2 and HIV-1 depends on the multiplicity of infection and the activity of the IFN-I system. Indeed, at a lower MOI, the relative advantage of TIPs decreases and the same effect takes place for higher IFN-I levels. These features suggest that TIP therapy could be less effective under ART-suppressed conditions when combined with IFN stimulation.”
- Lines 554-557 mention recombination risk. Please expand on the potential consequences based on known HIV molecular biology.
As the reviewer suggested earlier, the heterodimerization allows recombination thus arising a safety risk as HIV could recombine with TIP genomes, potentially restoring lost functions or generating new variants. We have added this statement to the above mentioned text.
- Add a paragraph of limitations: single-cell model, parameter assumptions, homogeneous cell population, lack of adaptive immunity modeling, etc.
In response to this and similar earlier suggestions, we have added to Discussion the text highlighting various aspects of limitations of the model.
- The conclusion should be more concise and forward-looking. Instead of repeating details, highlight the main finding, therapeutic implication (TIPs as immune sensitizers or combination partners, not monotherapy), contribution of the model (rational design tool), and an outlook toward cure.
Reviewer 1 suggested to rewrite or delete Conclusion section since this section is not mandatory according to the journal guideline. We have already extended the description of the main findings in the Discussion. In response to the suggestion of the Reviewer, we have now added to the end of Discussion the following statement.
“The mathematical model can be used as (i) a rational design tool for conducting in vitro experiments with TIPs as combination partners for suppressing HIV-1 replication in various target cells and (ii) to hypothesis on the modalities of combination therapies incorporating TIPs.”
Minor
- A thorough revision of the English is strongly recommended. There are numerous grammatical and spelling errors that need to be corrected. I have noted some of them in my minor comments, but many more remain and should be addressed systematically.
We have edited many sentences in order to improve English.
- Replace “non-humate primates” with “non-human primates”
- Abstract: “The model predict” by “The model predicts”
- Abstract: “reduction of HIV-infection” by “reduction of HIV infection”
- Introduction: “defective interfering particle (TIPs)” by “defective interfering particles (TIPs)”
- L29: “HIV is dipliod” by “HIV is diploid”
- L43: Clarify “this interference” (currently ambiguous)
- L51: “…a parallel set of equations the TIP-2 replication…” by “…a parallel set of equations for the TIP-2 replication…”
- L150: “In HIV-1 life cycle, he encoded messenger RNAs…” by “In the HIV-1 life cycle, the encoded messenger RNAs…”
- L449: “…cell incrasese with…” by “…cell increases with…”
- L450: “However. the extent…” by “However, the extent…”
All minor remarks (24-33) have been addressed. All indicated typos and mistakes have been corrected.
- While the abbreviation list is useful, ensure all acronyms (MOI, ISGs, ODE, MC, IFN-I in the abstract, etc.) are defined at first use.
The list of abbreviations can be found at the end of the paper. Nevertheless, we additionally define every term just after its first appearance.

Round 2
Reviewer 2 Report
Comments and Suggestions for Authors
The authors have addressed my questions and comments.
Author Response
We thank the Reviewer for insightful comments and the thorough work on our manuscript.
We hope that we have improved our English enough to publish an article in the Journal.
Reviewer 3 Report
Comments and Suggestions for Authors
Thank you for addressing my comments; the manuscript has improved considerably. My main and most pressing concern is that the manuscript requires a professional and comprehensive English language review. Numerous grammatical and spelling errors remain, which weaken the quality of your work. Please ensure that the entire text is carefully proofread by a professional editor.
Comments on the Quality of English LanguageL37: “HIV-1 is dipliod…” by “HIV-1 is diploid…”
L48: Replace “non-humate primates” by “non-human primates”
L72: “…enabling to reduce the basic reproduction rate…” by “…enabling the reduction of the basic reproduction rate…” or “…that enables the reduction…”
L90-91: “…responses of ISGs (Interferon-stimulated genes). in the infected cell.” Full stop followed by lowercase i.
L102: “…representing the Tat-Rev-dependent regulation mRNA abundance…” by “…representing the Tat- and Rev-dependent regulation of mRNA abundance…”
L317: “In this algorithm, the dynamics for any component, xn, automatically switches…” by “…automatically switch…”
L416: “…The variation of the finally secreted vision…” by “…The variation in the final number of secreted virions…”
L488: “…over-predicts the amount of secreted heterozygous particles, Specifically…” by “…over-predicts the amount of secreted heterozygous particles; specifically, the maximal output…”
Author Response
We thank the Reviewer for insightful comments on the language of our manuscript.
L37: “HIV-1 is dipliod…” by “HIV-1 is diploid…”
L48: Replace “non-humate primates” by “non-human primates”
L72: “…enabling to reduce the basic reproduction rate…” by “…enabling the reduction of the basic reproduction rate…” or “…that enables the reduction…”
L90-91: “…responses of ISGs (Interferon-stimulated genes). in the infected cell.” Full stop followed by lowercase i.
L102: “…representing the Tat-Rev-dependent regulation mRNA abundance…” by “…representing the Tat- and Rev-dependent regulation of mRNA abundance…”
L317: “In this algorithm, the dynamics for any component, xn, automatically switches…” by “…automatically switch…”
L416: “…The variation of the finally secreted vision…” by “…The variation in the final number of secreted virions…”
L488: “…over-predicts the amount of secreted heterozygous particles, Specifically…” by “…over-predicts the amount of secreted heterozygous particles; specifically, the maximal output…”
All indicated remarks have been corrected.
We hope we have improved the English language enough to publish the paper in the Journal.
